# Detection and prevalence of SARS-CoV-2 co-infections during the Omicron variant circulation in France

Antonin Bal [1,2,3,4], Bruno Simon[1,2,4], Gregory Destras[1,2,3,4], Richard Chalvignac[1,2], Quentin Semanas[1,2], Antoine Oblette[1,2], Grégory Quéromès[1,3], Remi Fanget[1], Hadrien Regue[1,2], Florence Morfin[1,2,3], Martine Valette[1], Bruno Lina [1,2,3] & Laurence Josset[1,2,3] ✉

From December 2021-February 2022, an intense and unprecedented co-circulation of SARS-CoV-2 variants with high genetic diversity raised the question of possible co-infections between variants and how to detect them. Using 11 mixes of Delta:Omicron isolates at different ratios, we evaluated the performance of 4 different sets of primers used for whole-genome sequencing and developed an unbiased bioinformatics method for the detection of co-infections involving genetically distinct SARS-CoV-2 lineages. Applied on 21,387 samples collected between December 6, 2021 to February 27, 2022 from random genomic surveillance in France, we detected 53 co-infections between different lineages. The prevalence of Delta and Omicron (BA.1) co-infections and Omicron lineages BA.1 and BA.2 co-infections were estimated at 0.18% and 0.26%, respectively. Among 6,242 hospitalized patients, the intensive care unit (ICU) admission rates were 1.64%, 4.81% and 15.38% in Omicron, Delta and Delta/Omicron patients, respectively. No BA.1/BA.2 co-infections were reported among ICU admitted patients. Among the 53 co-infected patients, a total of 21 patients (39.6%) were not vaccinated. Although SARS-CoV-2 co-infections were rare in this study, their proper detection is crucial to evaluate their clinical impact and the risk of the emergence of potential recombinants.

Since the first SARS-CoV-2 genome was published in January 2020, five variants of concern (VOC), characterized by increased transmissibility and/or immune escape capacity, have circulated worldwide[1]. Omicron (lineage B.1.1.529*), the last VOC to date, was first detected in South Africa[2] in November 2021 and displaced the previously circulating Delta (lineage B.1.617.2*) VOC by February 2022. While the Delta and Omicron variants share a number of mutations along their genome, Omicron is characterized by a large number of specific mutations, predominantly in the S gene[3]. In December 2021, a sub-lineage of the

Omicron variant, named BA.2, was detected and co-circulated with the Omicron BA.1 lineage from January 2022. Although derived from a common ancestor (lineage B.1.1.529), BA.1 and BA.2 are highly divergent with 27 specific mutations for the latter[3].

In France, the fifth wave of the SARS-CoV-2 epidemic was characterized by a sustained co-circulation of the Delta and Omicron (lineage BA.1) variants from November 2021-January 2022. Lineage BA.2 was first detected in France in late December 2021, and its proportion has since increased linearly[4]. Thus, the unprecedented

[1]Laboratoire de Virologie, Institut des Agents Infectieux, Laboratoire associé au Centre National de Référence des virus des infections respiratoires, Hospices Civils de Lyon, F-69004 Lyon, France. [2]GenEPII sequencing platform, Institut des Agents Infectieux, Hospices Civils de Lyon, F-69004 Lyon, France. [3]CIRI, Centre International de Recherche en Infectiologie, Team VirPath, Univ Lyon, Inserm,U1111, Université Claude Bernard Lyon 1, CNRS, UMR5308, ENS de Lyon, F-69007 Lyon, France. [4]These authors contributed equally: Antonin Bal, Bruno Simon, Gregory Destras. ✉e-mail: laurence.josset@chu-lyon.fr

sustained co-circulation of genetically divergent lineages observed from November 2021 to February 2022 may have been suitable for co-infections with a risk of subsequent recombination events. Few cases of Delta and Omicron variant co-infections have been reported recently, but a systematic assessment of the prevalence of SARS-CoV-2 co-infections, including BA.1/BA.2 co-infections, has not been explored on a large data set[5–7]. As Omicron and Delta have been associated with distinct COVID-19 severity, the clinical spectrum of these co-infections also needs to be determined[8,9].

Detection of SARS-CoV-2 co-infection using whole-genome sequencing (WGS) is not trivial and can be hampered by several problems: i) uneven performances of primer sets used to amplify SARS-CoV-2 genome with possible amplification bias of some genomic regions of specific variants[10,11]; ii) possible sample contamination during the sequencing process which requires independent validation on duplicate extracts[12,13]; and iii) lack of unbiased and validated bioinformatics methods able to systematically detect co-infections. Previously published reports on SARS-CoV-2 Delta/Omicron co-infections were based on manually curated lists of divergent mutations and visual examination of their relative frequencies along the genome[5–7]. De-novo assembly methods have also been described to assemble different viral genomes present in one sample[14] but are computationally intensive.

Herein, we used different mixed ratios of Delta and Omicron cell culture isolates in order to assess the performances of four different primer sets for the detection of Delta/Omicron co-infections. A co-infection score was determined to warn about probable co-infection. The prevalence of Delta/Omicron and BA.1/BA.2 co-infections were then estimated on a large data set of sequences obtained from random surveillance of out-patients and systematic sequencing of hospitalized patients.

## Results

### Evaluation of 4 different primer sets to detect SARS-CoV-2 co-infections using WGS

To simulate co-infections, Delta (B.1.617.2) and Omicron (BA.1) viral isolates were mixed using different Delta:Omicron ratios: 0:100, 10:90, 20:80, 30:70, 40:60, 50:50, 60:40, 70:30, 80:20, 90:10, and 100:0. Resulting mixes had fixed viral loads (median = 4.2 log10 cp/ml; IQR = 0.4, Supplementary Data 1). Four sets of primers (Artic V4 and V4.1, Midnight V1 and V2) were used in duplicate on extracts to test the impact of PCR amplification prior to sequencing on co-infection characterization. All mixes were sequenced to 1 M paired-end reads, leading to SARS-CoV-2 genome covered > 98% with a median coverage of 2276X (IQR = 315X) (Supplementary Data 1).

The evaluation of the primer sets was performed using a previously published method based on a curated list of mutations specific to Delta and to Omicron derived from co-variants[5–7] (Supplementary Data 1 and Fig. 1). More than 90% of the Delta-specific mutations were found in all mixes with the 4 primer sets. In contrast, the detection rate of Omicron-specific mutations ranged from 26% in the 90:10 mix using Midnight primers (V1 and V2) to >77% for mixes with the expected frequency of Omicron above 30% (Fig. 1A and Supplementary Data 1). Lower detection rate of Omicron-specific mutations was associated with different primer bias preferentially amplifying Delta over Omicron (Suppl Results, Supplementary Data 2). Medians of covered allele frequency for the specific mutations were used to estimate viral frequency. Relations between measured and expected frequency were not linear (Fig. 1B). Over-estimation of Delta was observed for all mixes with all primer sets, and especially in mixes with expected frequency of Delta under 30% and sequenced with Midnight primer sets. Measured frequencies of Delta for the 10:90 mix were between 30–33% with Midnight V1 and V2, and 21–25% with Artic V4 and V4.1.

Importantly, consensus sequence calling based on majority rule resulted in artefactual chimeric Delta-Omicron sequences for several mixes and with different patterns depending on the primers used for

amplification (Fig. S4). Sequences bearing both Delta- and Omicron-specific polymorphisms were found independently of the bioinformatic pipeline used to call the consensus sequence (Fig. S5). Omicron sequences bearing Delta-specific mutations were found in mixes with Delta expected frequency of 10–30%. The highest number of Delta-specific mutations was observed in the 20:80 mixes sequenced with Midnight primers, and in the 30:70 mixes sequenced with Artic primers (Fig. S5), which were the mixes with 50% measured frequency of Delta (Fig. 1B). Such sequences were observed with the four primer sets in all duplicates only for the 20:80 mix (Fig. 1C). With Artic primer sets, chimeric sequences were characterized by Omicron sequences bearing the S:L452R and M:I82T mutations, in relation with amplicon 76 and 89 bias (Suppl Results), respectively, and additional Delta-specific mutations with increasing Delta concentration. With Midnight primers, chimeric sequences were characterized by Omicron sequences with 3' end of the genome belonging to Delta (starting from nt 27,874, in relation with amplicon 28 bias).

Altogether, the Artic V4.1 primers were the least biased for Delta/Omicron co-infection detection and relative frequency estimation, but all primer sets could lead to artefactual chimeric sequences, highlighting the importance of proper co-infection detection.

### Novel bioinformatic algorithm to detect SARS-CoV-2 co-infections using WGS

Independent to this specific set of mutations, an agnostic approach was developed to detect co-infections regardless of the lineage present in the sample, as long as these lineages are genetically distinct (Fig. 2). This approach is based on the identification of a potential secondary lineage, after excluding variants shared with the main lineage. A secondary lineage is potentially identified when six or more specific mutations of this lineage are present as minor variants; this threshold was determined on composite criteria based on the mixes and samples as detailed in supplementary methods. Two ratios are calculated: the main lineage mutation ratio and the secondary lineage mutation ratio quantifying the fraction of present mutations among covered lineage-defining and lineage-specific mutations, respectively. Based on this co-infection detection script, co-infection was successfully identified in all mixes, except for the pure Delta (100:0) and Omicron (0:100) isolates, for which only Delta (lineage B.1.617.2) and Omicron (lineage BA.1) were identified as the main lineages, respectively. B.1.617.2 and BA.1 were identified as the main and secondary lineages, respectively, in all mixes with expected frequency of Delta above 40%, independent of the primer sets (Fig. 2 and Supplementary Data 1). Main lineage mutation ratios were above 0.9 for all mixes (Fig. 2A). Secondary lineage mutation ratios were between 0.216 and 0.941 (Fig. 2B). The lowest ratios were found for the mix 90:10 with only 0.23, 0.25, 0.39, and 0.55 of BA.1-specific mutations found using Midnight V2, Midnight V1, Artic V4, and Artic V4.1, respectively. These low ratios were associated with primer bias preferentially amplifying Delta over Omicron (Suppl Results, Supplementary Data 2).

To test whether our pipeline may detect co-infection in mixes with a lower percentage of Delta at risk of leading to chimeric sequences, we performed additional mixes with Delta: Omicron ratios of 1:99 and 5:95 that were sequenced in 10 replicates (Fig. S6 and Supplementary Data 3). All chimeric sequences were detected by seqmet as co-infected, except one chimeric 1:99 mix sequenced with Midnight V1 primers classified as non co-infected (Suppl Results, Fig. S6).

Altogether, the results of the unbiased co-infection detection scripts were consistent with the curated list approach with a better detection of low abundant BA.1 with Artic primers.

### Prevalence of SARS-CoV-2 co-infections during the fifth wave in France

Between December 6, 2021 (week 49-2021) and February 27, 2022 (week 08-2022), 23,242 samples were sequenced using Artic V4 or V4.1

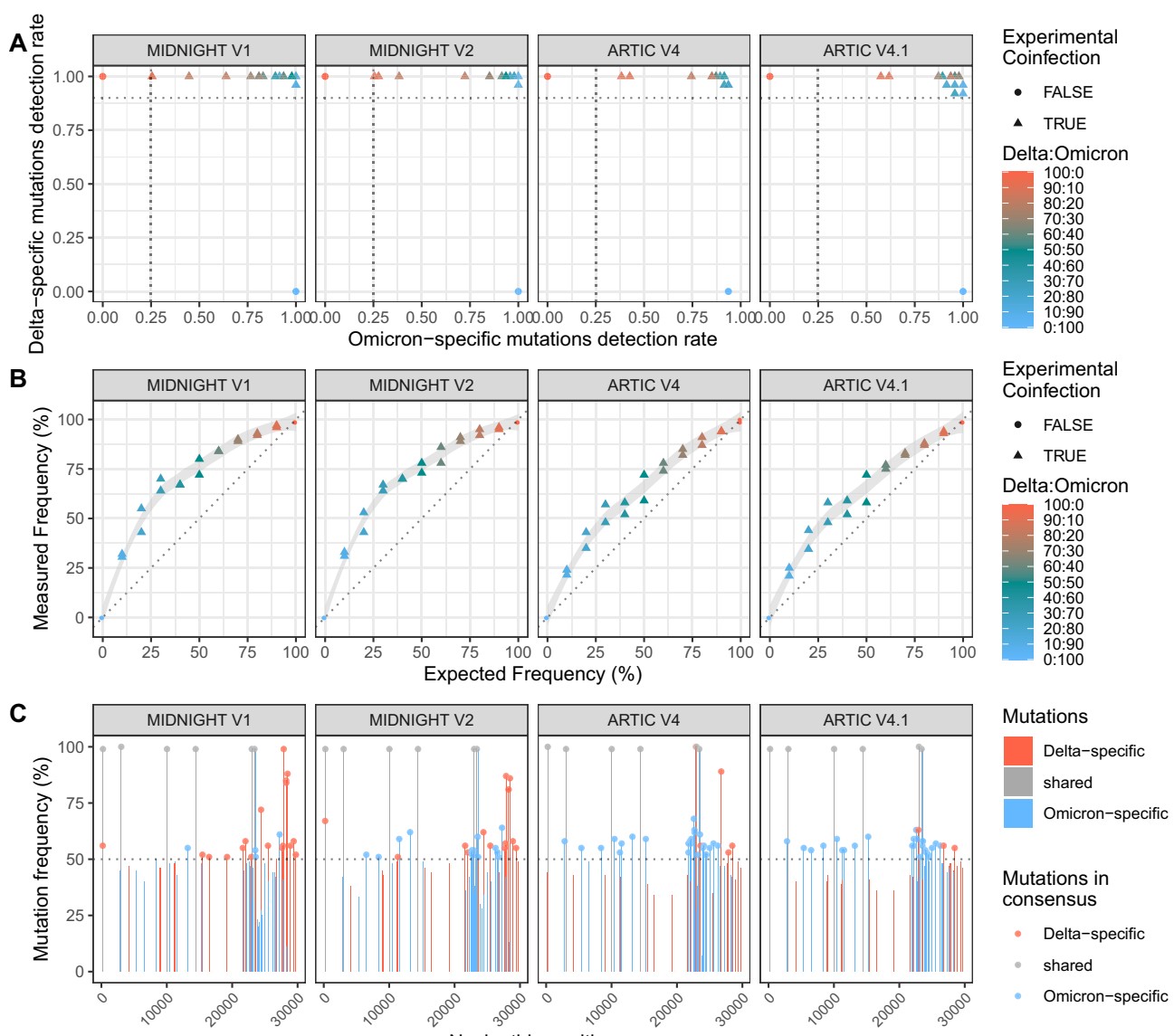

**Fig. 1 | Evaluation of 4 primer sets for WGS of Delta:Omicron mixes: Midnight V1, Midnight V2, ARTIC V4 and ARTIC V4.1.** Eleven mixes of Delta: Omicron isolates with different proportions were extracted and sequenced in duplicate. **A** Detection rate of Delta- and Omicron-specific mutations. These detection rates are defined as the number of Delta- or Omicron-specific variants found in each sample (as minor or major allele) out of the total number of Delta- or Omicron-specific variants based on covariants.org. Horizontal line (Delta-specific mutation ratio) at 0.9 and vertical line (Omicron-specific mutation ratio) at 0.25 discriminate co-infections from pure isolates. **B** Frequency of the Delta variant was measured using the median of covered allele frequencies of Delta-specific mutations and compared with the expected relative frequency in each mix. Pure isolates (denoted as 100:0 for Delta and 0:100 for Omicron) are depicted with dots, and Delta: Omicron mixes with triangles. Colors represent the expected frequency of each mix using a red (Delta: 100:0) to blue (Omicron: 0:100) gradient. **C** Representation of variant allele frequency of Delta- and Omicron-specific mutations for the mix 20:80. Delta-specific mutations are depicted in red, Omicron-specific mutations in blue and shared mutations are in grey. Horizontal lines at 50% show which mutations are called in the consensus sequence based on the majority rule. Chimeric sequences with both Delta and Omicron mutations were detected for the four primer sets.

primer sets as they became available. In total, WGS (coverage >90%) were obtained for 21,387 samples collected from Flash surveys (n = 16,220) and from HCL and peripheral hospitals (n = 5,167).

Among the 21,387 samples, 61 samples (0.29%) had a secondary lineage identified with a positive secondary lineage mutation ratio (Fig. 3A). Notably, changing the specific mutation cutoff between 3 and 17 had no significant impact on this detection rate (Suppl Results, Fig. S7). To rule out potential contamination during initial sequencing, all these 61 samples were re-extracted and sequenced in duplicate. In total, eight samples had no secondary lineage identified in duplicate while sequenced with high efficiency (coverage >90%), suggesting potential contamination during the first sequencing process (Suppl Results, Figs. S8 and S9). In contrast, 53 samples had a positive

secondary lineage mutation ratio in duplicate: 28 samples were identified as a Delta/Omicron (BA.1) co-infection; 1 sample was identified as a Delta/Omicron (BA.2) co-infection; 24 samples were identified as a co-infection between two different Omicron lineages (BA.1 and BA. 2) (Fig. 3A).

For these 53 natural co-infections, relative abundance of minor lineages were correlated in duplicate (Pearson correlation coefficient = 0.92, p-value <2.2e-16) with a median relative abundance of 20% (iqr = 26.25) (Fig. 3B). Minor lineages were identified as the secondary lineages in both duplicates for 45 samples but were identified either as the main or secondary lineage in duplicates for 8 samples with a higher relative abundance of minor lineage (median relative abundance = 38,75%, Kruskall–Wallis p-value = 9.479e-06) (Fig. 3B).

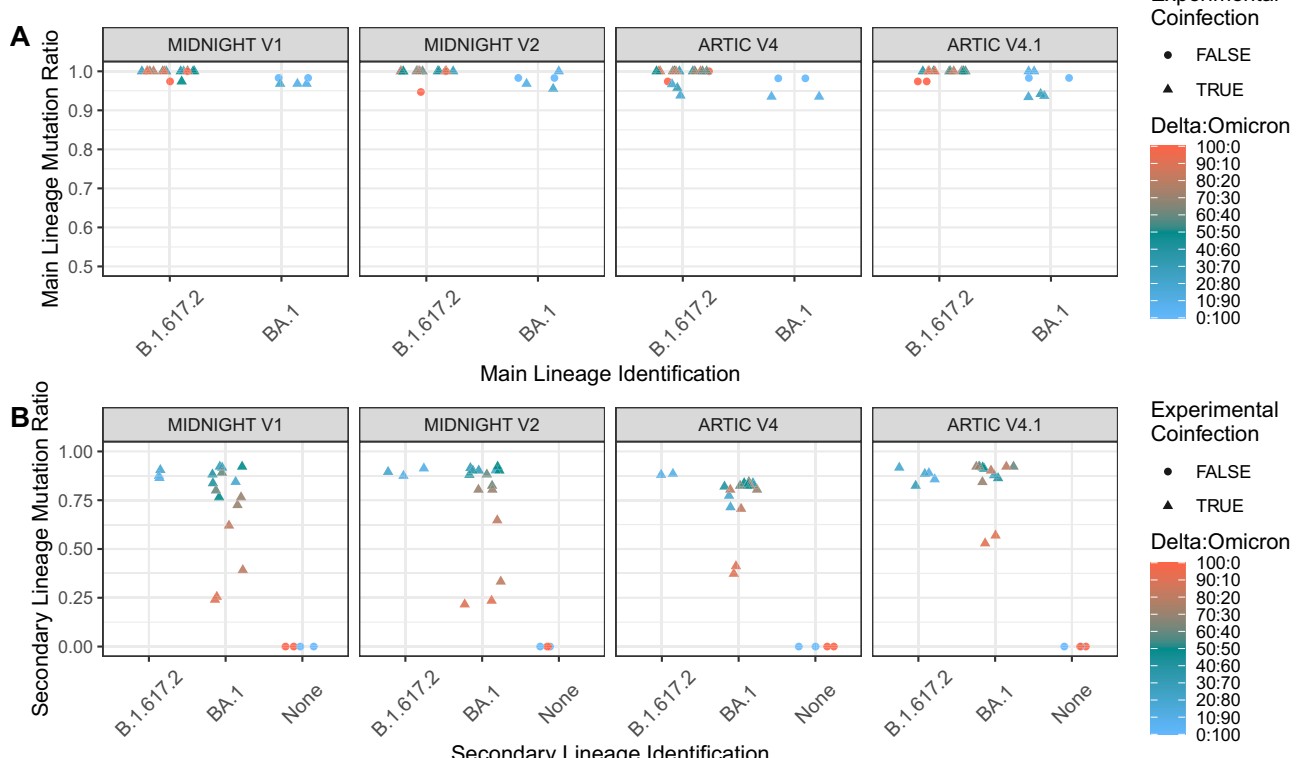

**Fig. 2 | Best main and secondary lineage matches identified for each Delta:
Omicron mix, based on without a priori comparison to a comprehensive
lineage mutation database. A** Best main lineage match ratio for each Delta:
Omicron mix, considering all mutations expected for the given lineage. **B** Best
secondary lineage match ratio for each mix, considering mutations expected for
the given lineage, excluding those in common with the main lineage. Pure isolates
(denoted as 100:0 for Delta and 0:100 for Omicron) are depicted with dots, and
Delta: Omicron mixes with triangles. Colors represent the expected frequency of
each mix using a red (Delta: 100:0) to blue (Omicron: 0:100) gradient.

The number of mutations specific to the minor lineage present in
the consensus sequence was used as a proxy to identify chimeric
sequences (Fig. 3C). Chimeric sequences were identified in 69/106
(65%) natural co-infected duplicates. All samples with a minor lineage
above 38% were chimeric with a median of 4 minor lineage-specific
mutations present in the consensus sequence (Fig. 3C).

All co-infections were confirmed by visual examination of vcf
plots using the lists of specific mutations from our lineage variant
database (Figs. S10–S12). Uniform frequencies of lineage-specific
mutations along the genome were observed in each sample, except
for three samples: "021228537801 (R1)" and "021229656701 (R3)"
among the Delta/BA.1 co-infections and "722000801801 (R2)" among
the BA.1/BA.2 co-infections (Suppl Results, Fig S13). In these samples,
the distribution of lineage-specific allele frequencies suggests the
presence of potential recombinants with different relative abundance
within co-infected samples (Suppl Results, Figs. S14 and S15). The
suspicion of a recombination event among co-infected samples was
therefore established for 3 out of 53 samples (5.6%).

Delta/Omicron (BA.1) co-infections were detected between weeks
50-2021 and 04-2022 (Fig. 4). Considering only the period of Delta and
Omicron (BA.1) co-circulation at relative frequencies above 1% (Weeks
50-2021 to 05-2022), Delta/Omicron (BA.1) prevalence was estimated
at 0.18% (28/15,253; 95 CI: 0.12–0.26% assuming a binomial distribu-
tion). The highest prevalence of Delta/Omicron (BA.1) co-infections
was observed in weeks 51 and 52-2021 with 0.31% (6/1,921) and 0.25%
(7/2,847), respectively. These 2 weeks were characterized by the
highest co-circulation of Delta and Omicron (BA.1) (week 51: 69.2%
Delta and 30.1% BA.1; week 52: 34.5% Delta and 65.1% BA.1). BA.1/BA.2
co-infections were detected between weeks 04 and 08-2022 with
prevalence reaching 0.78% of the sequences during the highest co-

circulation of BA.1 and BA.2 (week 08-2022: 57.3% BA.1 and 42% BA.2).
Considering only the period of BA.1 and BA.2 co-circulation at relative
frequencies above 1% (Weeks 03 to 08-2022), BA.1/BA.2 prevalence was
estimated at 0.26% (24/9,120; 95 CI: 0.17–0.39%).

**Clinical presentation of Delta/Omicron and BA.1/BA.2 co-
infections**
To assess the impact of co-infections on clinical presentations,
demographic features, including age and sex were reported for 13,187
out-patients, 6242 hospitalized patients, and 803 healthcare workers
(Table 1). In the three groups, no significant difference was noted
between BA.1 and BA.2 infections regarding proportion of men
($p > 0.05$ Fisher tests) or regarding median age for out-patients and
healthcare workers (p > 0.05, Mann–Whitney test). Therefore, BA.1 and
BA.2 cases were grouped into Omicron cases for further analysis. Delta
cases were significantly older than Omicron cases for out-patients
($p = 0.003$, Mann–Whitney test) and for hospitalized patients
($p < 0.001$, Mann–Whitney test). Delta cases were also significantly
more predominant in men for hospitalized patients ($p < 0.001$, Fisher
test). No difference regarding age or sex was found between Delta and
Omicron cases for healthcare workers ($p > 0.05$, Mann–Whitney test or
the Fisher tests). Among the three groups, no significant difference in
age or sex was found for Delta/Omicron or BA.1/BA.2 co-infections
compared to any other type of infection ($p > 0.05$, Mann–Whitney test
or the Fisher tests), except for BA.1/BA.2 co-infections significantly
reported in younger out-patients than Delta out-patients ($p = 0.03$,
Mann-Withney test) and in younger hospitalized patients than Delta/
Omicron hospitalized patients ($p = 0.004$, Mann-Withney test). ICU
admission for hospitalized patients was reported in 5.56%, 15.38%,
1.95%, and 0% of Delta, Delta/Omicron, Omicron, and BA.1/BA.2 cases,

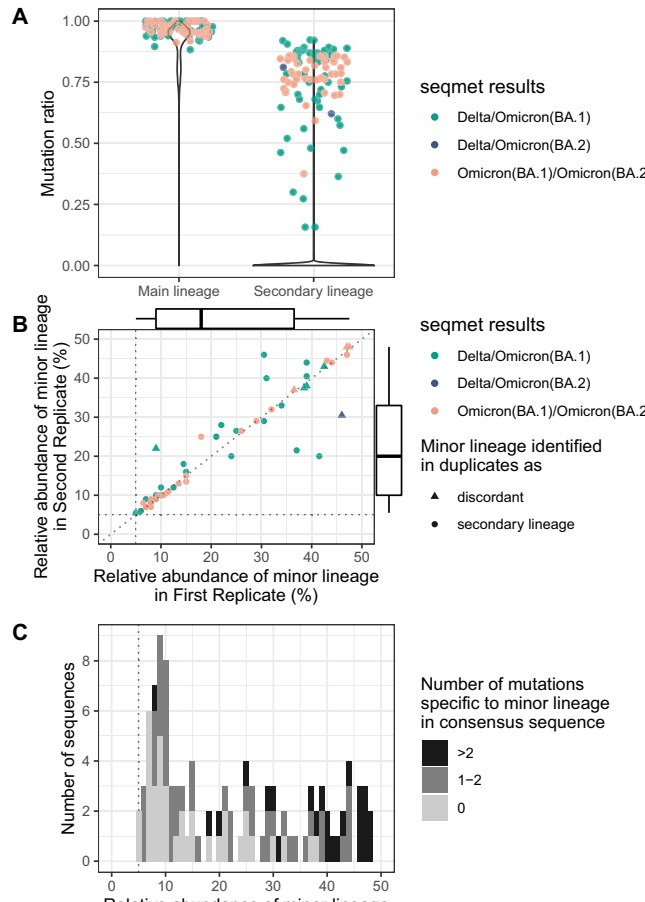

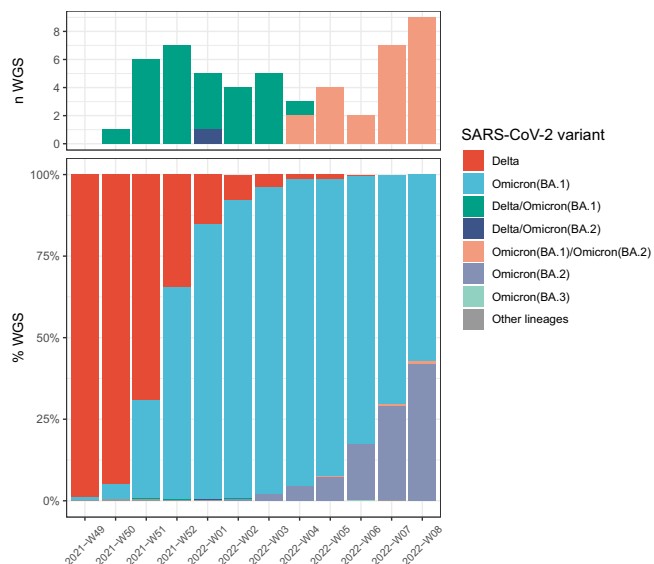

**Fig. 4 | Number of Delta/Omicron (BA.1), Delta/Omicron (BA.2) and BA.1/BA.2 co-infections during Delta, Omicron (BA.1), and Omicron (BA.2) co-circulation in France.** Colors indicate lineages: Delta (B.1.617.2 and AY.* lineages) in red, Omicron (BA.1) in blue, Omicron (BA.2) in purple, Omicron (BA.3) in light blue, other lineages (including B.1.640.1) in grey. Co-infections between Delta/BA.1 are in green, Delta/BA.2 in dark blue, and BA.1/BA.2 in salmon.

**Fig. 3 | Natural co-infections detected between December 6, 2021 and February 27, 2022. A** Violin plot of main and secondary virus mutation ratios for the 21,387 samples, including 61 samples sequenced in duplicate. The lineage ratio is considered to be 0 in cases where less than 6 mutations were found or covered with at least 100x for any lineage. Positive secondary virus mutation ratios in independent duplicates with concordant virus identification were considered as potential co-infections and depicted with dots ($n = 53$). Colors indicate lineages: co-infections between Delta/BA.1 are in green, Delta/BA.2 in dark blue, and BA.1/BA.2 in salmon. **B** Correlation between the relative abundance of the minor lineage in $n = 53$ natural co-infections sequenced in duplicate. Relative abundance is measured by the median of allele frequencies specific to the minor lineage. Distributions of relative abundance in first and second replicates are depicted as marginal boxplots. Boxplot represents median and IQR, with upper and lower whiskers extending to the largest and smallest value, respectively. Dotted lines are drawn at 5%, which is the threshold used to call a mutation in the variant calling step of seqmet. Colors indicate lineage as in panel **A**. Shapes indicate whether the minor lineage is identified as the secondary or main lineage in duplicate or a potential discordance between duplicates. **C** Number of chimeric sequences detected among the 53 natural co-infections sequenced in duplicate, depending on the relative abundance of the minor lineage. Chimeric sequences were defined by sequences bearing specific mutations of two lineages, with a gradient of color depending on the number of specific mutations from the less abundant lineage: 0 mutation (light grey), 1–2 specific mutations (medium grey), >2 specific mutations (dark grey).

respectively. Binomial logistic regression was tested and after correction by age and sex, ICU admission was significantly associated with Delta ($p < 0.001$, Wald Chi-square test) or Delta/Omicron (p < 0.01, Wald Chi-square test) infection. Among the 53 co-infected patients, a total of 21 patients (39.6%) were not vaccinated, corresponding to 9/16 (56.3%) hospitalized patients and 12/36 (33.3%) out-patients with no significant difference between the two groups ($p > 0.05$). Vaccinated patients (32/53, 60.4%), had received mainly two doses (18/32, 56.3%)

or three doses (13/32, 40.6%) of the vaccine. The median (IQR) delay between last dose and the date of RT-PCR test was 124 (57–166) days.

## Discussion

The intense co-circulation of distinct SARS-CoV-2 lineages in France since late 2021 gives the unique opportunity to assess the prevalence of Delta/Omicron and BA.1/BA.2 co-infections on a large data set of samples. Herein we found a prevalence of 0.18% and 0.26% for Delta/BA.1 and BA.1/BA.2 co-infections, respectively.

The estimation of the SARS-CoV-2 co-infection prevalence had been performed in several studies before the emergence of SARS-CoV-2 VOCs and led to conflicting findings. Liu and colleagues reported about 5% of co-infections in the United Arab Emirates between May and June 2020[15]. In the United Kingdom, Tonkin-Hill et al. found a co-infection in approximately 3–4% of samples collected in March and April 2020[16] and another study conducted in Brazil over a longer period (from May 2020 to April 2021) found a prevalence of 0.61%[17]. The low genetic diversity of SARS-CoV-2 observed during the pre-VOC period, the differences of genomic epidemiology depending on the geographical location, as well as the lack of standardization between the methods used may limit the interpretation of these results.

More recently, a study conducted on out-patients in the USA during the same period than the present study (November 2021 to February 2022) identified 20 Delta/Omicron co-infections out of 16,386 samples sequenced. The resulting prevalence of about 0.1% for Delta/Omicron is highly consistent with the prevalence found herein. However, no BA.1/BA.2 co-infection was reported by the authors as they specifically looked for Delta/Omicron co-infections using a curated list of mutations[7]. In contrast, the unbiased co-infection detection script presented herein and validated on mixed Delta: Omicron isolates was defined to detect likely co-infections with genetically divergent lineages, without a priori on these lineages. Furthermore, rates of detection for Omicron-specific mutations were below 50% in several Delta: Omicron mixes in relation to different primer bias preferentially amplifying Delta over Omicron. Thus, the threshold of alternate allele

**Table 1 | Clinical and demographic data of out-patients, hospitalized patients, and healthcare workers infected with Delta, Omicron (BA.1 or BA.2 lineages), or co-infected with Delta/BA.1 or BA.1/BA.2**

| | Out-patients | | | | | Hospitalized | | | | | Healthcare workers | | | | |
|---|---|---|---|---|---|---|---|---|---|---|---|---|---|---|---|
| | Delta (N = 4307) | Delta/Omicron (N = 15) | Omicron (N = 8828) | BA.1/BA.2 (N = 21) | Other lineages (N = 16) | Delta (N = 1601) | Delta/Omicron (N = 13) | Omicron (N = 4622) | BA.1/BA.2 (N = 3) | Other lineages (N = 3) | Delta (N = 103) | Delta/Omicron (N = 1) | Omicron (N = 699) | BA.1/BA.2 (N = 0) | Other lineages (N = 0) |
| **Age** | | | | | | | | | | | | | | | |
| Median (IQR) | 39.36 (24.01, 52.26) | 31.13 (19.94, 36.09) | 36.93 (22.23, 51.85) | 22.36 (16.66, 30.29) | 36.00 (27.25, 62.54) | 52.57 (33.36, 73.10) | 66.61 (50.79, 73.20) | 43.86 (26.73, 72.86) | 29.03 (28.33, 30.95) | 53.16 (39.23, 72.17) | 37.33 (28.16, 44.24) | 25.10 (25.10, 25.10) | 32.50 (25.74, 44.87) | NA | NA |
| **Sex** | | | | | | | | | | | | | | | |
| Male n (%) | 1,990 (46.20) | 10 (66.67) | 4,035 (45.71) | 6 (28.57) | 4 (25.00) | 852 (53.22) | 7 (53.85) | 2,084 (45.09) | 2 (66.67) | 1 (33.33) | 30 (29.13) | 0 (0.00) | 195 (27.90) | NA | NA |
| **Severity** | | | | | | | | | | | | | | | |
| ICU admission (%) | NA | NA | NA | NA | NA | 89 (5.56%) | 2 (15.38%) | 90 (1.95%) | 0 (0.00%) | 0 (0.00%) | NA | NA | NA | NA | NA |

fractions set at 85% by Bolze et al., can miss some co-infection cases and underestimate their prevalence.

With SARS-CoV-2 genetic diversity increasing, amplicon primer sets may lead to drop-out[18] and preferential amplification of one lineage over the other in a co-infected sample. Indeed, updates of Midnight and Artic V4 primers into Midnight V2 and Artic V4.1, respectively, were designed to resolve specific Omicron drop-outs. In this study, Artic primers were less biased than Midnight primers for the estimation of viral frequencies in experimental mixes, and V4.1 had the best performance in detecting Omicron-specific mutations in mixes with low proportions of Omicron isolates. However, none of the 4 primer sets tested avoided artefactual chimeric sequences in several mixes. Therefore, the detection of co-infections is important to avoid depositing such chimeric sequences in public repositories. To date (2022-03-18), 2,253 Omicron sequences and 1,156 Delta sequences have been tagged as "Multiple Lineage" in GISAID, including Omicron sequences bearing S: L452R + M: I82T, and may represent Delta/Omicron co-infections, contaminations or genuine recombinants.

The co-infection detection methodology presented herein highlights a key step in WGS data analysis that could be transposed for any pathogens analyzed by mapping on a single reference. The amount of work required to adapt the approach depends on the ploidy of the pathogen, its genetic diversity, and whether a pre-existing clade/lineage definition and a well-curated database of the currently circulating strains exist. Therefore, applying this approach to influenza virus, monkeypox, or respiratory syncytial viruses, for example, would be fairly straight-forward but would require experimental testing to adjust the lineage-specific mutation cutoff to the laboratory settings (level of contamination background) and the genetic distance between co-circulating lineages.

The present study brings the first insight into the clinical severity of patients presenting with a co-infection. While Delta/Omicron co-infections were associated with higher rates of ICU admission compared with Omicron infections, no severe forms were noticed in BA.1/BA.2 co-infections. These distinct clinical spectrums need to be further investigated to include the study of vaccination status between co-infected patients and non-co-infected patients. Herein, we found that 60.4% of co-infected patients were vaccinated, while the overall vaccination coverage during this period was above 90% in France[19]. A strict comparison could not be performed as it required to stratify the vaccination coverage by clinical status, which was not available for all patients at a national level. The clinical characteristics of potential recombinants resulting from a co-infection event[20] should also be assessed on a large data set. Among the 53 cases of co-infection identified in the present study, three possible novel recombinants occurring in co-infected samples may have been detected: two potential Delta-BA.1 recombinant with unbalanced frequencies of Delta- and Omicron-specific mutations and one potential BA.1-BA.2 recombinant with unbalanced frequencies of BA.1- and BA.2 specific mutations. Virus isolation in cell culture was possible only for one of these recombinants, as the two others samples had unfortunately been inactivated. The rate of recombination event during co-infection reported in the present study was similar to the one reported by, Bolze et al., who identified a Delta-BA.1 recombinant population in 1 out of 20 co-infected cases in the USA[7].

This study has several limitations. First, co-infections with a minor lineage <5% or with genetically closed lineages (less than 6 specific mutations) would not be detected with our pipeline. Second, the follow-up duration for the assessment of BA.1/BA.2 co-infection is short, and as BA.2 detection is still increasing in France, the prevalence of BA.1/BA.2 co-infection may be underestimated. In addition, ICU admission and vaccination status information was lacking for some Flash cases, limiting the clinical characterization of co-infected patients. Finally, additional sequencing methods were not tested, such as metagenomics or hybrid capture, which are less prone to

amplification bias toward specific lineages[7]. However, these techniques have lower sensitivity and lower throughput and were therefore not suitable as first-line sequencing methods in our laboratory[21].

In conclusion, our findings emphasize the importance of using appropriate experimental and bioinformatic methods for the comprehensive identification of SARS-CoV-2 co-infections. Although these events are rare, SARS-CoV-2 co-infections need to be properly identified as they can lead to the emergence of new variants after a recombination event.

## Methods

### Mix of Delta: Omicron cell culture isolates
The Delta and Omicron variants were isolated in cell culture from nasopharyngeal swabs (NPS). Following interim biosafety guidelines established by WHO, NPS were inoculated on confluent Vero E6 TMPRSS2 cells with DMEM supplemented with 2% penicillin–streptomycin, 1% L-glutamine, 2% G418, and 2% inactivated fetal bovine serum. Plates were incubated at 37 °C with 5% $CO_2$ for 48 h. The cytopathic effects were monitored daily; samples were harvested when positive. Viral isolates were quantified using RT-PCR https://www.who.int/docs/default-source/coronaviruse/real-time-rt-pcr-assays-for-the-detection-of-sars-cov-2-institut-pasteur-paris.pdf and sequenced to confirm the lineage and the absence of low-frequency diversity. The Delta and Omicron isolates were then diluted to reach similar viral loads (Ct = 19) and mixed using different Delta: Omicron ratios: 0:100, 10:90, 20:80, 30:70, 40:60, 50:50, 60:40, 70:30, 80:20, 90:10, and 100:0. After nucleic acid extraction performed in duplicate, all RNA extracts were diluted ten-fold and stored in several aliquots under the same conditions (frozen at −80 °C). Thus, all extracts were subjected to one freeze-thaw cycle for all sequencing methods. To determine the detection limit of our pipeline, an additional experiment was performed in the same conditions, mixing Delta: Omicron at 1:99, 5:95, and 10:90.

### SARS-CoV-2 sequencing
RNA from culture supernatants was extracted in duplicate using the automated MGISP-960 workstation using MGI Easy Magnetic Beads Virus DNA/RNA Extraction Kit (MGI Tech, Marupe, Latvia) for the first extract and EMAG platform (bioMérieux, Lyon, France) for the second extract. A total of four sets of SARS-CoV-2 primers were tested on duplicate extracts of culture supernatants. The primers used were Artic V4 NCOV-2019 Panel (IDT, ref #100,10008554 μM), Artic V4.1 NCOV-2019 Panel (IDT, ref # 100,10011442 μM), SARS-Cov2-Midnight-1200 V1 (IDT, ref # 100,10007184 μM) and SARS-Cov2-Midnight-1200 V2 (IDT, ref # 100,10007184 μM with equimolar spike of SARS-CoV_1200_28_LEFT_27837T: TTTGTGCTTTTTAGCCTTTCTGtT), at 10 μM final concentration.

For Artic V4 and V4.1 primers, cDNA synthesis and amplification were performed using the COVIDSeq-Test™ (Illumina, San Diego, USA). For Midnight V1 and V2 primers, cDNA synthesis and amplification were performed using LunaScript® and Q5® Hot Start (New England Biolabs, Ipswich, USA), respectively. Libraries were prepared with the COVIDSeq-Test (Illumina, San Diego, USA), and samples were sequenced with 100 bp paired-end reads using the NovaSeq 6000 Sequencing system SP flow cell.

The routine SARS-CoV-2 sequencing protocol in our laboratory is based on COVIDSeq-Test™ (Illumina, San Diego, USA) using Artic V4 or V4.1 primers as they become available.

### SARS-CoV-2 quantification
Quantitative viral load was determined for each mix with RT-qPCR using SARS-CoV-2 R-gene kit (bioMérieux, Lyon, France) with four quantification standards targeting the SARS-CoV-2 N gene: QS1 to QS4 respectively 2.5.10^6, 2.5.10^5, 2.5.10^4, 2.5.10^3 copies/mL of a SARS-CoV-2 DNA standard.

### Sample selection
The samples sequenced at the National Reference Center (NRC) of Respiratory Viruses of *Hospices Civils de Lyon* (HCL) were selected for this study i) from systematic sequencing of hospitalized patients in the Lyon area (university hospital of Lyon, HCL) and from HCL health care workers; ii) from random sequencing performed during the weekly Flash surveys conducted by the EMERGEN consortium (French consortium for the genomic surveillance of emerging pathogens). The Flash surveys are nationwide surveys where all private and public diagnostic laboratories in France are asked to provide to the NRC and other sequencing centers a fraction of positive samples from one day per week ranging from 25 to 100% depending on the number of positive cases detected at the national level[4,22]. The prevalence of SARS-CoV-2 co-infections was estimated on samples collected both in the HCL and in Flash samples sequenced by the NRC of HCL. To assess the clinical presentations of co-infected patients, three groups were selected: out-patients of Flash surveys, hospitalized patients of Flash surveys and HCL, and healthcare workers of HCL, excluding follow-up samples.

### Bioinformatics
Reads were processed using the in-house bioinformatic pipeline seqmet (available at https://github.com/genepii/seqmet). Paired reads were trimmed with cutadapt to remove sequencing adapters, and low-quality ends, only keeping reads longer than 30 bp[23]. Alignment to the SARS-CoV-2 reference genome MN908947 was performed by Minimap2[24]. Mapped reads were processed to remove duplicates tagged by picard, then realigned by abra2 to improve indel detection sensitivity and finally clipped with samtools ampliconclip to remove read ends containing primer sequences[25,26]. Variants present at frequencies of 5% or above were called using freebayes, decomposed and normalized with vt, and then filtered with bcftools to eliminate false positives[26–28]. To detect co-infection, obtained vcf files were compared to a lineage variant database by a script, both developed internally to this end. The database consists of vcf files listing variants found in 50% or more of 100 randomly selected sequences for a given pangolin lineage in the full GISAID data set available (extracted on 02 February 2022). The database is available at https://github.com/genepii/seqmet-db. The co-infection detection script searches for each lineage any major or minor variant matching expected variants of the putative main lineage and then searches for any minor variant matching any other lineage, excluding variants in common with the main lineage. Since the variants of the main lineage are excluded when searching the secondary lineage, variants for the secondary lineage are specific to this lineage, whereas variants for the main lineage are defining variants of the lineage. Variants were expected when they occurred on a position covered with at least 100 reads. This approach provides putative main and secondary lineages contained in sequenced sample reads, along with a ratio of the number of observed out-of-expected variants in each case (referred to as the main lineage mutation ratio and the secondary lineage mutation ratio). These ratios are used as primary criteria to identify which lineages are the closest to those in the sequenced sample reads; ties are broken based on raw counts of observed variants, which are expected higher than a minimum distance threshold. The minimal distance threshold depends on the genetic distance between co-infecting lineages and on the experimental settings of the sequencing lab, especially, the background level of contamination with in-house protocols and automation. It was therefore adjusted on these parameters during the period we analyzed (Suppl methods, Figs. S1–S3). Based on the training and real data presented here, this threshold was set to a minimum of 6 lineage-specific variants (Suppl methods, Fig S1 and Fig S2). The threshold was best suited to identify Delta/Omicron and Omicron/Omicron co-infections, but not all Delta/Delta co-infections (Fig. S3). As lineages are identified based on mutation ratio, the main lineage is not necessarily

the most abundant and lineage identification could lack precision on the sub-lineage scale for which very close ratios can be found.

For Delta: Omicron mixes, vcf files were also analyzed using the curated list of clade-defining mutations of Delta and Omicron (lineage BA.1) as previously published[5,6]. This list is based on https://covariants.org/variants as of 11/02/2022, excluding 21846 C > T (S: T95I), which is present in 40% of Delta variants. These data are presented in the first paragraph of the results section.

## Statistical analysis

Continuous variables are presented as mean ± standard deviation (SD) or median with interquartile range (IQR) and compared using nonparametric Kruskal–Wallis or Mann–Whitney tests. Proportions were compared using the chi-squared or Fisher's exact test, as appropriate. The relationship between the ICU admission in relation to VOC infection, sex, and age, were examined with the binomial logistic regression technique. The significance was tested by the Wald Chi-square statistic. A $p$-value of <0.05 was regarded as statistically significant. Statistical analyses were conducted using R software, version 4.0.5 (R Foundation for Statistical Computing).

## Ethics

Samples used in this study were collected as part of an approved ongoing surveillance conducted by the NRC of HCL. The investigations were carried out in accordance with the General Data Protection Regulation (Regulation (EU) 2016/679 and Directive 95/46/EC) and the French data protection law (Law 78–17 on 06 January 1978 and Décret 2019–536 on 29 May 2019). Samples were collected for regular clinical management, with no additional samples for the purpose of this study. Patients were informed of the research and their non-objection approval was confirmed. This study was approved by the ethics committee of the *Hospices Civils de Lyon* (HCL), Lyon, France, and registered on the HCL database of RIPHN studies (AGORA N°41).

## Reporting summary

Further information on research design is available in the Nature Research Reporting Summary linked to this article.

## Data availability

The GISAID accession numbers of the Delta and Omicron virus isolates used for experimental mixes are EPI_ISL_11171170 and EPI_ISL_11171169, respectively. Sequencing data of the Delta: Omicron mixes were deposited on the SRA database under accession PRJNA817870 and PRJNA853723, and dehosted sequencing data of NPS with co-infections were deposited under accession PRJNA817806.

## Code availability

The seqmet bioinformatic pipeline is publicly available at https://github.com/genepii/seqmet[29]. The software versions are provided in Table S1. The '58a2c4d28288b54cba425225bdaa9a0d642048ca' commit of seqmet should be used to reproduce the results presented herein. In addition, all data and R scripts used to generate the results are publicly available at https://github.com/genepii/seqmet/tree/main/script/article.

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

## Acknowledgements

We would like to thank all the members of the GenEPII sequencing platform who contributed to this investigation. We also thank all the laboratories, clinicians, and patients involved in this work. We would like to especially thank Dr. Claire Bardel, Dr. Romain Hernu, Dr. Bénédicte Clément and Théophile Boyer for their input during the revisions. This work was carried out within the framework of the French consortium on surveillance and research on infections with emerging pathogens via microbial genomics (*consortium relatif à la surveillance et à la recherche sur les infections à pathogènes EMERgents* via *la GENomique microbienne EMERGEN*; https://www.santepubliquefrance.fr/dossiers/coronavirus-covid-19/consortium-emergen). We acknowledge the authors, originating and submitting laboratories of the sequences from GISAID's EpiCov Database. All submitters of data may be contacted directly via the GISAID website www.gisaid.org-*Santé publique France*, the French national public health agency. -*Caisse nationale d'assurance maladie (Cnam)*, the national health insurance funds. -"Enhancing Whole Genome Sequencing (WGS) and/or Reverse Transcription Polymerase Chain Reaction (RT-PCR) national infrastructures and capacities to respond to the COVID-19 pandemic in the European Union and European Economic Area" (Grant Agreement ECDC/HERA/2021/007 ECDC. 12221

## Author contributions

A.B., B.S., G.D., F.M., B.L., and L.J. conceived the study. R.C., A.O., and Q.S. performed the sample preparations and sequencing. B.S. and H.R. performed bioinformatic analysis. M.V. and R.F. performed virus isolation. A.B., G.D., G.Q., and L.J. were the main writers of the manuscript. All authors reviewed and approved the final version of the manuscript.

## Competing interests

The authors declare no competing interests.
