## [Peer Review File · Nature Communications]

Detection and prevalence of SARS-CoV-2 co-infections during the Omicron variant circulation in FranceREVIEWER COMMENTS

Reviewer #1 (Remarks to the Author):

The manuscript prepared by Bal et al. Describes the validation of a wet and dry lab SARS-CoV-2 genomic workflows to detect VOC co-infections. The authors describe the use of these techniques to examine the frequency of SARS-CoV-2 VOC co-infection in a large cohort of community and hospitalised patients with COVID infections in France.

The detection of SARS-CoV-2 co-infections has become increasingly important as recombinant SARS-CoV-2 lineages have been recognised internationally. The manuscript is well written and describes important parameters that need to be implemented and standardised internationally to effectively quantify and differentiate SARS-CoV-2 VOC co-infection and recombination. The manuscript also describes the frequency of SARS-CoV-2 co-infections in community-based populations and in cases with severe COVID-19 infections.

Main points. The authors described important bioinformatic parameters that are needed to accurately identify SARS-CoV-2 co-infections and provide a github repository for their code. Unfortunately, there is no content on this github page therefore I cannot review this pipeline.

Minor points.

Consider outlining dates in full within the abstract. The use of abbreviated and epi weeks is confusing.

Methods – SARS-CoV-2 sequencing section, please reference primer sets used and indicate the concentration of each primer in the multiplex reactions i.e are different concentrations of primers sets used.

Line 224. Please outline the reason that 11 samples, after repeat extraction, were no longer found to harbour co-infections.

Line 307. The two potential recombinant cases detected in the study are of interest. Could the authors highlight the likely breakpoints of these recombinants. Unfortunately the genome names and mutation labels in Figure S2 are too small. Potentially the recombinants could be highlighted in a separate diagram?

It is interesting that the authors highlight chimeric reads in samples a low percentage of one lineage. Could the authors outline the detection limits of their workflow.

Figure 1. The mutations listed along the x-axis are too small to read, consider an alternative labelling system.

Reviewer #2 (Remarks to the Author):

In the current study, Bal et al develop and apply a method to detect co-infection in SARS-CoV-2. The Authors combine proportional mixtures of Delta and Omicron viruses in vitro and show that they can detect mixed infections employing a novel bioinformatic pipeline applied to deep sequencing data. They then apply this pipeline to over 20,000 WGS samples from France collected during the co-circulation of Delta, BA.1 and BA.2 and reproducibly detect co-infections between these lineages. The work is novel, timely, important, of high quality and of broad interest and I support publication in Nature Communications. The novel pipeline has the potential to be used by others to detect co-infections. I have only minor comments and clarifications outlined below.

Minor comments

It doesn't seem that the code behind the pipeline is currently available. The Authors provide a link to <https://github.com/genepii/seqmet> but this is currently empty. It is therefore not currently

possible for the pipeline to be used by others or be assessed for robustness. Additionally, it is not clear what will be included in the available code and whether this will contain all of the code used to identify co-infections and produce plots similar to those in the manuscript.

The paper should better discuss the limitations of the newly developed pipeline. In the abstract (lines 28-29), it is claimed that the method “can detect all co-infections irrespective of the SARS-CoV-2 lineages involved”. This is not, however, the case. Many Pango lineages of SARS-CoV-2 differ by fewer than the six SNPs currently required by the pipeline to detect the presence of a minor lineage. In these cases, it will not be possible to detect co-infection. Additionally, clearly co-infections with the same lineage cannot be detected. This sentence and other references to being able to detect all co-infections should therefore be tempered.

Additionally, some clarifications on the method should be provided, specifically:

1. How is the ‘main lineage’ of the sample chosen by the pipeline? Is this the lineage with the largest number of supporting mutations? What would the pipeline choose in situations where multiple lineages have the same number of supporting mutations?
2. Why was six mutations chosen as the cut-off to identify a minor lineage? It’s unclear how increasing or decreasing this cut-off would influence the ability to detect co-infections as this will depend on the number of mutations separating different lineages. Some justification for this cut-off should be provided.
3. How would the pipeline proceed in cases where two lineages are separated by more than six SNPs but only a fraction of the SNPs are detected in the sample? Are all of the mutations associated with a lineage required to identify it?

Lines 28-29 – as described above, “all co-infections” should be rephrased

Lines 43-44 – it would be ideal to state the Pango lineages of these VOCs the first time they are mentioned for clarity: Delta is B.1.617.2* and Omicron is B.1.1.529*.

Lines 180-183 – it would be useful to present an analysis of the Omicron mutations that are missed in each sample. Is it always the same mutations that are missed or are the missed mutations sample-specific?

Line 183 – should this reference Table S2 rather than Table S1?

Lines 274-288 – Dezordi et al (2021, medRxiv, <https://www.medrxiv.org/content/medrxiv/early/2021/09/23/2021.09.18.21263755.full.pdf>) is another study that should be included here as this attempts to identify SARS-CoV-2 co-infections from >2000 samples in Brazil collected before May 2021

Lines 308-309 – these potential recombinant samples are interesting. I’m unclear, however, why the SNP frequencies of Delta and Omicron mutations are not at 100%. If these were cases of infection with a single Delta-Omicron recombinant virus, I would expect the mutations obtained from Delta and the mutations obtained from Omicron to be at 100% frequency. Please clarify.

Lines 314-315, “These data suggest that rates of recombination within co-infection hosts are low (<5%)” – it doesn’t seem possible to robustly infer a rate of recombination within co-infected hosts from the available data. If a patient is co-infected with two viruses, say Delta and Omicron, and these viruses recombine, the mutation frequencies of the variant-specific mutations will be intermediate, exactly as they would be in the absence of recombination. The presence or absence of recombination cannot therefore be inferred from SNP frequencies alone. Additionally, many viruses arising from recombination events are likely to remain at low frequency within an individual infection and may therefore not transmit onwards to be detected in another patient. Further, recombination cannot be easily detected when a patient is infected with a single virus or highly similar viruses, even if it is occurring very regularly. It therefore seems difficult to present a rate of recombination from the available data.

Lines 325-328 – is it possible to describe whether this method would be applicable to other pathogens and what would be required?

Lines 440-442 – are these colours meant to be included in the legend for figure 3? They don't seem to correspond to colours in that figure?

Figure 1A – I may be missing this but the ratio of Delta-specific mutations and the ratio of Omicron-specific mutations don't seem to be described anywhere which makes it difficult to understand exactly what is being plotted in this figure. It would be useful to describe what these terms mean.

Reviewer #3 (Remarks to the Author):

The manuscript entitled "Detection and prevalence of SARS-CoV-2 co-infections during the Omicron variant circulation, France, December 2021 - February 2022" by Bal et al. describes an automated data analysis pipeline for amplicon sequencing-based identification of co-infection with 2 of the 5 known variants of concern of SARS-CoV-2. The authors describe the publicly-available pipeline and validate its performance on a titrated mix of cultured Delta:Omicron variants. They then apply the pipeline to amplicon sequencing data from >20,000 patient samples positive for SARS-CoV-2 during December 2021 - February 2022 and identify 53 reproducibly detectable co-infections. The authors discuss the importance of identifying co-infections in an automated and scalable way, including to identify recombination events and limit description of artificially chimeric viral consensus genome sequences.

Major comments

- Line 190: "Importantly, consensus sequence calling based on majority rule resulted in artefactual chimeric Delta-Omicron sequences for several mixes and with different patterns depending on the primers used for amplification (Fig S1)" -> This is an important observation. Is this also true for other commonly used pipelines? Can the authors test? Did the authors quantify how often this was the case with natural co-infections?
- Line 205: "A secondary lineage is identified only if 6 specific mutations are present" -> How was this cutoff derived and optimized? Will this perform equally well for lineages with fewer lineage-specific mutations? Can the authors provide data?
- Line 225: "To rule out potential contamination during initial sequencing, all these 64 samples were re-extracted and sequenced in duplicate. In total, 53 samples had a positive secondary lineage mutation ratio in duplicate" -> Can the authors comment on results for the remaining 11 samples?
- Line 233: "Uniform frequencies of lineage-specific mutations" -> How was "uniform" defined and the determination made? Solely by manual inspection? Could the authors include a flag in the software or at least discuss how this could be done?
- Do the authors have access to information on the immune status of patients with co-infections?
- Fig 3: can the authors add a panel showing the distribution of median allele frequencies for the 'secondary virus' in natural co-infections? Using the information from Fig 1B, can the authors calculate the corrected variant ratio in the natural co-infection cases (Delta- Omicron)? what was the rate of chimeric consensus sequences in natural co-infections?

Minor comments

- Can the authors provide viral loads in copies rather than Ct? e.g., line 173 "Resulting mixes had fixed viral loads (median Ct = 22.8; IQR=1.5)"
- Line 214: "Secondary lineage mutation ratios were between 0.216 and 0.941 (Fig 2B)"; This seems surprising given the high depth of coverage (Line 176: "All mixes were sequenced to 1 M paired-end reads leading to SARS-CoV-2 genome covered > 98% with median coverage of 2276X (IQR=315X)" -> Can the authors discuss the low ratios for some of the cases?

- Line 224: "Among the 21,387 samples, 64 samples (0.30%) had a secondary lineage identified with positive ratios (Fig 3)" -> What is the definition of 'positive ratio'?
- Line 143: "For Delta:Omicron mixes, vcf files were also analyzed using the curated list of clade-defining mutations of Delta and Omicron (lineage BA.1) as previously published." -> Can the authors provide these results?
- Fig 1A/B and 2A/B -> The red-blue color gradient is lacking resolution making it hard to map colors to ratios of the virus mix-> Suggest 3-color gradient

Reviewer #1 (Remarks to the Author):

The manuscript prepared by Bal et al. Describes the validation of a wet and dry lab SARS-CoV-2 genomic workflows to detect VOC co-infections. The authors describe the use of these techniques to examine the frequency of SARS-CoV-2 VOC co-infection in a large cohort of community and hospitalised patients with COVID infections in France.

The detection of SARS-CoV-2 co-infections has become increasingly important as recombinant SARS-CoV-2 lineages have been recognised internationally. The manuscript is well written and describes important parameters that need to be implemented and standardised internationally to effectively quantify and differentiate SARS-CoV-2 VOC co-infection and recombination. The manuscript also describes the frequency of SARS-CoV-2 co-infections in community-based populations and in cases with severe COVID-19 infections.

Main points.

Q1. The authors described important bioinformatic parameters that are needed to accurately identify SARS-CoV-2 co-infections and provide a github repository for their code. Unfortunately, there is no content on this github page therefore I cannot review this pipeline.

A1. Unforeseen circumstances prevented us from uploading the pipeline described in the paper on schedule. It is now publicly available and will be maintained and kept up-to-date if there is an appeal for a more mainstream use. The analysis pipeline and database creation pipeline have been separated for practical reasons, and can be found respectively on (<https://github.com/genepii/seqmet>) and (<https://github.com/genepii/seqmet-db>). In order to verify the reproducibility of the results included in the paper, the '58a2c4d28288b54cba425225bdaa9a0d642048ca' commit of seqmet should be used. The parameters used for the analysis are included in the json parameters template of the pipeline 'seqmet/piperun/000000_seqmet_varcall_ncov/params_01.json'.

Minor points.

Q2. Consider outlining dates in full within the abstract. The use of abbreviated and epi weeks is confusing.

A2. The corrections have been made accordingly.

Q3. Methods – SARS-CoV-2 sequencing section, please reference primer sets used and indicate the concentration of each primer in the multiplex reactions i.e are different concentrations of primers sets used.

A3. The following primers and concentrations were used:

Primers Artic V4 NCOV-2019 Panel (IDT, ref #10008554, 100µM), Artic V4.1 NCOV-2019 Panel (IDT, ref # 10011442, 100µM), SARS-Cov2-Midnight-1200, (IDT, ref # 10007184, 100µM) and SARS-Cov2-Midnight-1200 V2 (IDT, ref # 10007184, 100µM, Equimolar spike of SARSCoV_1200_28_LEFT_27837T: TTTGTGCTTTTTAGCCTTTCTGtT) were used for multiplex reactions at 10µM final concentration.

We clarified this point in the revised version of the manuscript (lines 98-104 in the version with tracked changes).

Q4. Line 224. Please outline the reason that 11 samples, after repeat extraction, were no longer found to harbour co-infections.

A4. Among the 11 samples, 3 samples were mistakenly included in the initial version of the manuscript as they did not reach the 90% genome coverage threshold in the second passage and are therefore not part of the 21,387 samples with WGS analyzed in the manuscript. Only 61 samples had >90% genome coverage in both duplicates, including 8 that had discordant secondary lineage mutation ratios.

Additional analyses were performed for these 8 samples (**Fig S8** and **Fig S9**) and are now described in the **Supplementary results** reproduced below:

Investigation of samples with discordant secondary lineage mutation ratios in duplicates

Among 61 samples with positive secondary lineage mutation ratios in the first passage, 8 samples had null ratios in the second passage, thus no longer found to harbour co-infections. These samples were characterized by lower secondary lineage mutation ratios in the first replicate than confirmed co-infected samples (median ratio of 0.38 vs 0.78, Kruskal-Wallis p-value = 0.0083) (**Fig S8A**) and lower relative abundance of the

minor lineage in the first replicate (median relative abundance of 9.5% vs 18%, Kruskal-Wallis p-value = 0.03) (**Fig S8 B**). The distribution of minor alleles along the genome in duplicate suggests that a contamination arose during the first sequencing process, either during RNA extraction, PCR amplification or library preparation (**Fig S9**). For $\frac{5}{8}$ samples, minority variants from the secondary lineage were present all along the genome, suggesting that the contamination arose most likely during the RNA extraction. For $\frac{3}{8}$ samples, minority variants were present only on 2 to 3 amplicons, suggesting contamination may have arisen after PCR amplification with a limited number of amplicons. Additionally, for 4 samples, as secondary lineage-specific mutation relative frequencies were $\sim 5\%$, we cannot rule out that a co-infection could have been present, but allele frequencies were not reproducible around 5%.

These results are referenced in the main text (lines 280-283) and detailed in the **Supplementary results, Fig S8** and **Fig S9**.

Q5. Line 307. The two potential recombinant cases detected in the study are of interest. Could the authors highlight the likely breakpoints of these recombinants. Unfortunately the genome names and mutation labels in Figure S2 are too small. Potentially the recombinants could be highlighted in a separate diagram?

A5. As suggested, we have added a dedicated figure (**Fig S14**) for the recombinants in order to highlight the likely breakpoints. Please note that following the suggestion Q4 of Reviewer 3, a third potential recombinant at low frequency was detected and is now presented in the main and supplementary results.

We have also added new results in supplementary materials of the revised version of the manuscript to further explore the recombinants. These data are reproduced below:

Investigation of possible recombinants

Three possible recombinants occurring in co-infected samples may have been detected in this study. Virus isolation in cell culture was possible only for one of these

recombinants (021228537801), as the two others samples had unfortunately been inactivated in the originating laboratories after sampling.

For the sample with a BA.1/BA.2 co-infection (sample # 722000801801), a BA.1-predominant region was defined from nt 0 to 6,512 where the relative frequencies of BA.1-specific mutations were above 70%, while relative frequencies of BA.2-specific mutations were around 6% (**Fig S14B**). From nt 8,393 frequencies were reversed and BA.2 was predominant except for two positions (nt 22,686 and nt 22,673). Thus, this sample could contain a BA.1-BA.2 recombinant at 58% relative abundance with a likely breakpoint located between nt 6,512 nt and 8,393. Due to low viral load (Ct = 30), we could not conduct more investigations for this sample. Of note, this low viral load may be associated with poor reproducibility regarding variant allele frequencies.

Two potential recombinants were identified among Delta/Omicron co-infections (samples # 021228537801 and 021229656701). For 021228537801, a AY.43-predominant region was defined from nt 0 to 11,332, while BA.1.1 was predominant from nt 15,240 to 29,909 except for one position in one replicate (nt 22,917) (**Fig S14A**). The high viral load of this sample (Ct=22) enabled further investigations. First, we tested a long-read sequencing approach based on amplification with Midnight V2 primers, a ligation-based library preparation (PCR Barcoding Kit without fragmentation, Oxford Nanopore Technologies, ONT) and a sequencing on GridION platform with Q10 chemistry and FLO-MIN106 flow cell (ONT). Despite a read length of 1200 pb, no sufficient number of reads overlapping specific AY.43 and BA.1.1-specific mutations could be produced. This approach was therefore not contributive to specify the position of the breakpoint. Then, we tried to isolate the potential recombinant using viral cell culture Vero E6 TMPRSS2 cells with two successive passages (P1 and P2). WGS with Artic V4.1 primers was then performed on the two culture supernatants.

A decrease of mixed populations throughout the passages was noticed (**Fig S14A**). In the P2 viral isolate, the frequencies of AY.43-specific mutations reached 100% and BA.1.1 specific mutations were no longer detected from nt 0 to 15,240. From nt 21,762, AY.43-specific mutations were no longer detected and frequencies of BA.1.1-specific mutations reached 100%. From nt 15,240 to 19,220, both AY.43 and BA.1.1-specific mutations were still detected with median frequencies of 35% and 65%, respectively. These results suggest that two potential recombinants have been selected after the two passages in cell culture, the main one with a potential breakpoint around nt 15,000 and a final relative abundance of 65% and the minor one with a potential breakpoint around nt 20,000 and a final relative abundance of 35%. To explore this, the P2 viral isolate processed with Artic V4.1 primer was analyzed with a different bioinformatics pipeline (<https://github.com/connor-lab/ncov2019-artic-nf>). P2 was also processed with a metagenomics (mNGS) method based on Revelo RNA-Seq High Sensitivity kit (Tecan, Männedorf, Switzerland), Illumina sequencing and mapping-based bioinformatics analyses (seqmet pipeline, <https://github.com/genepii/seqmet>). Both approaches showed similar results to those obtained with V4.1 primer followed by seqmet analyses that ruled out sequencing artefacts in this region of the genome (**Fig S15**). Of note, the consensus sequence generated from the second passage was unclassified with Pango v.4.1.1, suggesting that these recombinants have not been significantly detected.

Finally, the investigation of 021229656701 revealed a potential recombinant at 13% relative abundance since an increase and a decrease of 13% were found respectively for the allele frequencies of specific mutations defining the minor and the major lineage after the position 23,604 and from 23,948, the two positions defining the suspected region for the recombinant breakpoint (**Fig S14C**). It is worth noting that contrary to the

other recombinants discussed above, specific mutations of the minor lineage did not reach 50% frequency. The consensus sequence was thus not impacted.

Q6. It is interesting that the authors highlight chimeric reads in samples a low percentage of one lineage. Could the authors outline the detection limits of their workflow.

A6. We thank the reviewer for this comment.

In our manuscript, we described chimeric consensus sequences (and not chimeric reads) that were found in mixes with 10 to 50% Omicron. **Fig1C** highlighted the 80:20 mix (20% of Omicron) for which chimeric sequences were found with all primers in duplicates. Delta-specific mutations found in the consensus sequences were related to primer bias with some primers amplifying preferentially Delta over Omicron. Primer biases are now described in **Supplementary results** and **Table S3**.

In addition, we have performed additional experiments to address the detection limit question. These data have been added in the **Supplementary Results** and are reproduced below:

Detection limit of the seqmet pipeline

To test the detection limit of our pipeline to detect co-infections in mixes with lower percentage of Omicron that may also lead to chimeric consensus sequences, we performed additional mixes with Delta:Omicron ratios of 1:99 and 5:95 that were sequenced in 10 replicates (**Fig S6** and **Table S4**). Chimeric sequences were defined as sequences with both Delta- and Omicron-specific mutations in the consensus. For the 5:95 mixes, chimeric sequences were found in 10/10, 9/10, 10/10 and 6/10 samples sequenced with Midnight V1, Midnight V2, Artic V4 and Artic V4.1, respectively. All the 5:95 mixes had a positive secondary lineage mutation ratio and were thus identified as co-infections. In contrast, for the 1:99 mixes, the seqmet pipeline identified a co-infection in 9/10, 9/10, 7/10 and 7/10 samples, while 7/10, 1/10, 10/10, and 0/10 were chimeric sequences with Midnight V1, Midnight V2, Artic V4 and Artic V4.1, respectively. Therefore, for mixes with only 1% of Omicron, our workflow was not able to detect the co-infection in all replicates, while one of the 1:99 mix sequenced with

Midnight V1 was detected as not co-infected but led to chimeric sequences. This detection limit was expected as the threshold used for variant calling is 5%, therefore, in theory, only minor viruses above 5% should be detected. However, because of primer bias preferentially amplifying Delta over Omicron, in the 1:99 mixes, Delta had a measured relative abundance above 5%.

These new results are described in the main text (lines 263-267), in **Supplementary results** and supplementary materials (**Fig S6, Table S4**).

Q7. Figure 1. The mutations listed along the x-axis are too small to read, consider an alternative labelling system.

A7. **Fig 1C** was updated to include a more readable x-axis label (nucleotide position). Names of the mutations can be found in **Fig S4** by zooming in on each dot (vector images are provided).

Reviewer #2 (Remarks to the Author):

In the current study, Bal et al develop and apply a method to detect co-infection in SARS-CoV-2. The Authors combine proportional mixtures of Delta and Omicron viruses in vitro and show that they can detect mixed infections employing a novel bioinformatic pipeline applied to deep sequencing data. They then apply this pipeline to over 20,000 WGS samples from France collected during the co-circulation of Delta, BA.1 and BA.2 and reproducibly detect co-infections between these lineages. The work is novel, timely, important, of high quality and of broad interest and I support publication in Nature Communications. The novel pipeline has the potential to be used by others to detect co-infections. I have only minor comments and clarifications outlined below.

Q1. It doesn't seem that the code behind the pipeline is currently available. The Authors provide a link to <https://github.com/genepii/seqmet> but this is currently empty. It is therefore not currently possible for the pipeline to be used by others or be assessed for robustness. Additionally, it is not clear what will be included in the available code and whether this will contain all of the code used to identify co-infections and produce plots similar to those in the manuscript.

A1. Unforeseen circumstances prevented us from uploading the pipeline described in the paper on schedule. It is now publicly available and will be maintained and kept up-to-date if there is an appeal for a more mainstream use. The analysis pipeline and database creation pipeline have been separated for practical reasons, and can be found respectively on (<https://github.com/genepii/seqmet>) and (<https://github.com/genepii/seqmet-db>). In order to verify the reproducibility of the results included in the paper, the '58a2c4d28288b54cba425225bdaa9a0d642048ca' commit of seqmet should be used. The parameters used for the analysis are included in the json parameters template of the pipeline 'seqmet/piperun/000000_seqmet_varcall_ncov/params_01.json'

Data source and all R scripts used to produce the figures and supplementary materials are now provided in <https://github.com/genepii/seqmet/script/>.

Q2. The paper should better discuss the limitations of the newly developed pipeline. In the abstract (lines 28-29), it is claimed that the method "can detect all co-infections irrespective of

the SARS-CoV-2 lineages involved”. This is not, however, the case. Many Pango lineages of SARS-CoV-2 differ by fewer than the six SNPs currently required by the pipeline to detect the presence of a minor lineage. In these cases, it will not be possible to detect co-infection. Additionally, clearly co-infections with the same lineage cannot be detected. This sentence and other references to being able to detect all co-infections should therefore be tempered.

A2. We thank the reviewer for this comment.

We agree with the reviewer that our approach is not able to “detect all co-infections irrespective of the SARS-CoV-2 lineages involved” as initially erroneously claimed but only the lineages that differ by more than 6 specific-mutations. This sentence in the abstract was changed to “we developed an unbiased bioinformatics method for the detection of co-infections involving genetically distinct SARS-CoV-2 lineages” (line 29). We have also changed other references to this point, lines 244-245 and 382-383.

We have also added a section and a figure (**Fig S3**) in the supplementary methods to describe which lineages can be detected using the 6 mutations cut-off. This limitation has been added in the main method (lines 164-166) and in the discussion (lines 430-432).

Additionally, some clarifications on the method should be provided, specifically:

Q3. 1. How is the ‘main lineage’ of the sample chosen by the pipeline? Is this the lineage with the largest number of supporting mutations? What would the pipeline choose in situations where multiple lineages have the same number of supporting mutations?

A3. The main lineage of a sample is the lineage with the highest found/expected ratio of defining mutations. At least 6 defining mutations are required to define the main lineage. The aim of this approach is to determine the mutations that should be excluded when seeking a secondary lineage. Thus, the main lineage is not necessarily the most abundant, as highlighted in **Fig 3B** and discussed lines 288-292 and 166-168. In the case where multiple lineages have the same ratio of supporting mutations, the one with the higher found mutation count is chosen, and if there is still a tie, the lineage is chosen in reverse alphabetical order (longest sub-lineage in most cases).

The description of the co-infection detection algorithm has been detailed in **Supplementary methods** reproduced below:

Description of the co-infection detection algorithm

The co-infection detection consists in several steps based on a sample vcf file, a sample position-based depth bed file, a target region bed file and a lineage profile database generated with our database building pipeline (<https://github.com/genepii/seqmet-db>). Sample files are generated by bedtools and freebayes with our analysis pipeline, the target region bed file is provided with the pipeline and corresponds to the whole coding region in our case (<https://github.com/genepii/seqmet>). For a given sample, our pipeline first compares, thanks to a script, the sample vcf to each lineage vcf of the database to find a main lineage in the sample. Note that this main lineage is the one obtaining the best ratio of found out of expected variants, which is not necessarily the one with the highest allele frequency. All defining variants of the tested lineage for which the sample had a sequencing position depth higher than 100 reads are expected to be found if the tested lineage corresponds to one of the lineages contained in the sample. Then, each of these expected variants are searched in the sample vcf file considering both minor and major variants. A ratio of the number of found variants on the expected list is calculated and used to select, among the lineages for which the found variant count is higher than a cutoff, the best main lineage candidate. In the case where multiple lineages have the same ratio of supporting mutations, the one with the higher found mutation count is chosen, and if there is still a tie, the lineage is chosen in reverse alphabetical order (longest sub-lineage in most cases). A second round of this script is performed with two key differences. The defining variants of the newly found main lineage are excluded from the search, and only minor variants of the sample vcf file are compared to the tested lineage profiles. Performing this second round while excluding the main lineage-defining variants allows the search to be based only on variants specific to the putative secondary lineage when compared with the main lineage. Finally, the main and

secondary lineage, the found and expected variants, and the calculated ratio are reported in a summary file.

We have also added this information in the main manuscript (lines 153-159).

Q4. 2. Why was six mutations chosen as the cut-off to identify a minor lineage? It's unclear how increasing or decreasing this cut-off would influence the ability to detect co-infections as this will depend on the number of mutations separating different lineages. Some justification for this cut-off should be provided.

A4. As suggested, we have added 3 sections in supplementary methods and results to explain how the 6 specific mutation cutoff was derived, its impact on the detection of the different lineages, and on our results. These data are supported by 4 new supplementary figures (**Fig S1, S2, S3 and S7**) and are referenced in the main text (lines 159-164 and 277-279).

These supplementary data are reproduced below:

Determination of the optimal cutoff value for detecting the secondary lineage

The appropriate cutoff was determined empirically while training the pipeline on real data and positive controls included in each run. Based on the artificial Delta:Omicron mixes and all positive controls (cell culture isolates) sequenced during the study period (n=1084, including 892 pure cell culture isolates and 192 experimental mixes) a ROC curve was generated with the Youden index indicating a putative optimal point for this cutoff (**Fig S1A**). While the Youden index's best score corresponds to a minimum of 4 lineage-specific variants cutoff (TPR-FPR=0.952), the chosen cutoff of 6 while designing the study had a close score (TPR-FPR=0.932) and permitted the exclusion of false positives identified in the real dataset. Indeed, out of 7 samples which had 4 or 5 secondary lineage-specific mutations at first passage, only 1 sample retained 5 secondary lineage-specific mutations at second passage (**Fig S2**). Moreover, in the real dataset, among samples with a non null secondary lineage-specific mutations count,

2143/2257 (94.9%) samples had a mutation count ranging from 1 to 5 as shown in **Fig S1B**, which was similar to the proportion of 454/477 (95.1%) found in the positive controls sequenced during the study period.

Impact of the cutoff value on different lineages detection

The number of defining mutations depends on the lineage with a median of 40 defining mutations per lineage, a minimum of 26 mutations for AY.9.2 and a maximum of 64 mutations for BA.2 (**Fig S3A**). For the main lineage definition, the 6 defining mutations threshold is very low compared to the number of defining mutations per lineage (**Fig S3A**). On the other hand, the 6 specific mutations threshold to find a secondary lineage limits our ability to detect co-infections between closely related lineages. Specifically, 12321/43472 (28%) Delta/Delta co-infections with different lineages cannot be detected using the 6 specific mutation threshold (**Fig S3B**). At the time of writing, there was only one sub-lineage of BA.1 defined, BA.1.1, and co-infections between BA.1 and BA.1.1 would not be detected either as BA.1.1 has only 3 specific mutations in addition to the defining mutations of BA.1.

Impact of the cutoff value on the prevalence of natural co-infections

The number of samples with a secondary lineage identified in their first replicate depends on the cutoff value used to determine the secondary lineage (**Fig S7**). The chosen cutoff was 6 specific mutations, resulting in 0.29% of potentially co-infected samples (95% CI: 0.22%-0.37%, assuming a binomial distribution). Using only 1 specific mutation as the cutoff would identify a potential co-infection in 10.3% of the samples. In contrast, a cutoff between 3 and 17 specific mutations would not significantly change the estimated prevalence of co-infection (Fisher's p-value >0.05) (**Fig S7**). Excluding 1 specific mutation, the estimated prevalence varied between

0.542% (for a cutoff of 2 specific mutations) and 0.00935% (for a cutoff of 44 defining mutations).

Additionally, users can easily change this cut-off to match their experimental settings (contamination background may vary depending on experimental procedures and automation) and when seeking co-infection by closer lineages. This consideration has been added in the main text (lines 404-408).

Q5. How would the pipeline proceed in cases where two lineages are separated by more than six SNPs but only a fraction of the SNPs are detected in the sample? Are all of the mutations associated with a lineage required to identify it?

A5. To identify a lineage as a potential co-infectant, at least 6 mutations need to be found (with our current chosen cut-off). When less than 6 mutations are found, in the case of a co-infection by close lineages or in the case where the sequencing depth is insufficient for example, the result would remain inconclusive. When 6 or more mutations are found, even if the found/expected ratio is low, the sample is flagged as a potential co-infection. The description of the co-infection detection algorithm has been detailed in **Supplementary methods** (in the section: *Description of the co-infection detection algorithm*).

Q6. Lines 28-29 – as described above, “all co-infections” should be rephrased.

A6. We modified the abstract as follows:

“we developed an unbiased bioinformatics method for the detection of co-infections involving genetically distinct SARS-CoV-2 lineages” (lines 28-30)

Q7. Lines 43-44 – it would be ideal to state the Pango lineages of these VOCs the first time they are mentioned for clarity: Delta is B.1.617.2* and Omicron is B.1.1.529*.

A7. We added Pango lineages of these VOC in lines 43 and 45.

Q8. Lines 180-183 – it would be useful to present an analysis of the Omicron mutations that

are missed in each sample. Is it always the same mutations that are missed or are the missed mutations sample-specific?

A8. In mixes with 10% expected frequency of Omicron, most mutations were missed because they did not reach 5 % relative frequency (threshold used in seqmet to call a variant) (**Table S3**). In addition, some mutations were missed in the other mixes with a primer-specific pattern. Omicron-specific mutations at positions 22813 (S:K417N), 22882 (S:N440K), 22898 (S:G446S) were missed with Artic V4 primers, while Delta-specific mutation at position 22917 (S:L452R) was amplified preferentially in Delta:Omicron mixes, in relation with amplicon 76 (nt 22677- 23028) drop-out in Omicron samples (<https://community.artic.network/t/sars-cov-2-v4-1-update-for-omicron-variant/342>). Artic V4 amplicon 89 (nt 26587-26956) also led to Delta preferential amplification: Omicron-specific mutation at position 26709 (M:A63T) was missed in mixes with up to 60% Omicron expected frequency, while Delta-specific mutation at position 26767 (M:I82T) was preferentially sequenced. Mutations comprised in the Midnight amplicons 24 (nt 23544 –24714) and 28 (nt 27808 – 28985) also had amplification bias with preferential amplifications of Delta over Omicron, as reported in Rockett et al, Nature Com, 2022. Omicron-specific mutations at positions 23854 (S:N764K), 23948 (S:D796Y), 24130 (S:N856K), 24424 (S:Q954H), 24469 (S:N969K), 24503 (S:L981F), 28311 (N:P13L) and 28361 (N:30Del) were missed in several low frequency Omicron mixes sequenced with midnight primers, while Delta-specific mutations at positions 27874 (non coding), 28247 (ORF8: 118Del), 28270 (non coding) and 28461 (N:D63G) were present at high relative frequency.

In mixes with a low expected frequency of Delta (10-20% for Midnight primers; 20-30% for Artic primers), these amplification biases resulted in chimeric sequences characterized by Omicron sequences bearing Delta-specific mutations in these regions.

We have now included these data in **Table S3** and discuss these results in the **Supplementary results** (section: *Primer bias*) and in the main text (lines 215-217).

Q9. Line 183 – should this reference Table S2 rather than Table S1?

A9. Thank you for pointing out this error, this has been changed in the manuscript.

Q10. Lines 274-288 – Dezordi et al (2021, medRxiv,

<https://www.medrxiv.org/content/medrxiv/early/2021/09/23/2021.09.18.21263755.full.pdf>) is another study that should be included here as this attempts to identify SARS-CoV-2 co-infections from >2000 samples in Brazil collected before May 2021

A10. Thank you for sharing this study. We have now included this paper in the discussion section as well as another study from the United Kingdom that found a co-infection by multiple lineages in approximately 3–4% of samples. Please see below lines 366 to 374 of the revised version of the manuscript:

“The estimation of the SARS-CoV-2 co-infection prevalence had been performed in several studies before the emergence of SARS-CoV-2 VOCs that led to conflicting findings. Liu and colleagues reported about 5% of co-infections in the United Arab Emirates between May and June 2020¹⁷. In the United Kingdom, Tonkin-Hill et al., found a co-infection in approximately 3-4% of samples collected in March and April 2020 and another study conducted in Brazil over a longer period (from May 2020 to April 2021) found a prevalence of 0.61%. The low genetic diversity of SARS-CoV-2 observed during the pre-VOC period, the differences of genomic epidemiology according to the geographical location, as well as the lack of standardization between the methods used may limit the interpretation of these results.”

Q11. Lines 308-309 – these potential recombinant samples are interesting. I’m unclear, however, why the SNP frequencies of Delta and Omicron mutations are not at 100%. If these were cases of infection with a single Delta-Omicron recombinant virus, I would expect the mutations obtained from Delta and the mutations obtained from Omicron to be at 100% frequency. Please clarify.

A11. As suggested, we have now clarified how the presence of a recombinant within a mixed population could explain the observed distribution of allele frequencies along the genome (**Supplementary results, Fig S14**). For each of the suspected recombinants, at least 3 different lineages might be present: the 2 originating co-infecting lineages, plus the recombinant. Therefore, at each position, the allele frequency is the result of the abundance of each of these 3 lineages. This explains why specific allele frequencies do not reach 100% as we would expect with a pure recombinant (that would not include the 2 originating viruses). We have drawn barplots with these 3 populations and their estimated relative abundance below each allele frequency representation to clarify these hypotheses (**Fig S14**).

For one of the two Delta/Omicron samples, the high viral load enabled us to perform viral cell culture. A decrease of mixed populations throughout the passages was noticed. In the second cell culture passage, the frequencies of Delta-specific mutations reached 100% with BA.1-specific mutations no longer detected from nt 0 to 11,332. Inversely, after nt 21,762, Delta-specific mutations were no longer detected and frequencies of BA.1-specific mutations reached 100%. These results suggest that the potential recombinant has been selected after the two passages in cell culture (**Fig S14**).

We have now described in more detail the investigation of these potential recombinants in **Supplementary results** and **Fig S14**, as reproduced below:

Investigation of possible recombinants

Three possible recombinants occurring in co-infected samples may have been detected in this study. Virus isolation in cell culture was possible only for one of these recombinants (021228537801), as the two others samples had unfortunately been inactivated in the originating laboratories after sampling.

For the sample with a BA.1/BA.2 co-infection (sample # 722000801801), a BA.1-predominant region was defined from nt 0 to 6,512 where the relative frequencies of BA.1-specific mutations were above 70%, while relative frequencies of BA.2-specific mutations were around 6% (**Fig S14B**). From nt 8,393 frequencies were reversed and BA.2 was predominant except for two positions (nt 22,686 and nt 22,673). Thus, this sample could contain a BA.1-BA.2 recombinant at 58% relative abundance with a likely breakpoint located between nt 6,512 nt and 8,393. Due to low viral load (Ct = 30), we could not conduct more investigations for this sample. Of note, this low viral load may be associated with poor reproducibility regarding variant allele frequencies.

Two potential recombinants were identified among Delta/Omicron co-infections (samples # 021228537801 and 021229656701). For 021228537801, a AY.43-predominant region was defined from nt 0 to 11,332, while BA.1.1 was predominant

from nt 15,240 to 29,909 except for one position in one replicate (nt 22,917) (**Fig S14A**). The high viral load of this sample (Ct=22) enabled further investigations. First, we tested a long-read sequencing approach based on amplification with Midnight V2 primers, a ligation-based library preparation (PCR Barcoding Kit without fragmentation, Oxford Nanopore Technologies, ONT) and a sequencing on GridION platform with Q10 chemistry and FLO-MIN106 flow cell (ONT). Despite a read length of 1200 pb, no sufficient number of reads overlapping specific AY.43 and BA.1.1-specific mutations could be produced. This approach was therefore not contributive to specify the position of the breakpoint. Then, we tried to isolate the potential recombinant using viral cell culture Vero E6 TMPRSS2 cells with two successive passages (P1 and P2). WGS with Artic V4.1 primers was then performed on the two culture supernatants. A decrease of mixed populations throughout the passages was noticed(**Fig S14A**). In the P2 viral isolate, the frequencies of AY.43-specific mutations reached 100% and BA.1.1 specific mutations were no longer detected from nt 0 to 15,240. From nt 21,762, AY.43-specific mutations were no longer detected and frequencies of BA.1.1-specific mutations reached 100%. From nt 15,240 to 19,220, both AY.43 and BA.1.1-specific mutations were still detected with median frequencies of 35% and 65%, respectively. These results suggest that two potential recombinants have been selected after the two passages in cell culture, the main one with a potential breakpoint around nt 15,000 and a final relative abundance of 65% and the minor one with a potential breakpoint around nt 20,000 and a final relative abundance of 35%. To explore this, the P2 viral isolate processed with Artic V4.1 primer was analyzed with a different bioinformatics pipeline (<https://github.com/connor-lab/ncov2019-artic-nf>). P2 was also processed with a metagenomics (mNGS) method based on Revelo RNA-Seq High Sensitivity kit (Tecan, Männedorf, Switzerland), Illumina sequencing and mapping-based bioinformatics

analyses (seqmet pipeline, <https://github.com/genepii/seqmet>). Both approaches showed similar results to those obtained with V4.1 primer followed by seqmet analyses that ruled out sequencing artefacts in this region of the genome (**Fig S15**). Of note, the consensus sequence generated from the second passage was unclassified with Pango v.4.1.1, suggesting that these recombinants have not been significantly detected.

Finally, the investigation of 021229656701 revealed a potential recombinant at 13% relative abundance since an increase and a decrease of 13% were found respectively for the allele frequencies of specific mutations defining the minor and the major lineage after the position 23,604 and from 23,948, the two positions defining the suspected region for the recombinant breakpoint (**Fig S14C**). It is worth noting that contrary to the other recombinants discussed above, specific mutations of the minor lineage did not reach 50% frequency. The consensus sequence was thus not impacted.

We are also referencing these data in the main results lines 302-306.

Q12. Lines 314-315, “These data suggest that rates of recombination within co-infection hosts are low (<5%)” – it doesn’t seem possible to robustly infer a rate of recombination within co-infected hosts from the available data. If a patient is co-infected with two viruses, say Delta and Omicron, and these viruses recombine, the mutation frequencies of the variant-specific mutations will be intermediate, exactly as they would be in the absence of recombination. The presence or absence of recombination cannot therefore be inferred from SNP frequencies alone. Additionally, many viruses arising from recombination events are likely to remain at low frequency within an individual infection and may therefore not transmit onwards to be detected in another patient. Further, recombination cannot be easily detected when a patient is infected with a single virus or highly similar viruses, even if it is occurring very regularly. It therefore seems difficult to present a rate of recombination from the available data.

A12. We agree with the reviewer that a low frequency recombinant within a coinfecting sample and recombination between highly similar viruses would not be detected using our pipeline, similarly to co-infections with a minor lineage <5% or with genetically closed lineages that are

not detected with our pipeline. However, while the median of all specific allele frequencies (used to estimate relative abundance of each lineage) cannot differentiate Delta and Omicron co-infections with or without recombination, the distribution of allele frequencies at each position along the genome can suggest a recombination event (as discussed in **Supplementary results** and **Fig S13** and **S14**) and as previously described in A11. However, validation of recombinants requires additional experiments such as long read sequencing with phasing and virus isolation in cell culture.

Another limitation to robustly estimate the rates of recombination is the lack of longitudinal data in these co-infected samples. Therefore we agree the exact rate of recombination in a coinfecting host is likely underestimated using the available data and we removed the sentence “These data suggest that rates of recombination within co-infection hosts are low (<5%)”.

Q13. Lines 325-328 – is it possible to describe whether this method would be applicable to other pathogens and what would be required?

A13. This method could be transposed for any pathogen analyzed by mapping on a single reference. The amount of work required to adapt the approach depends on:

- The ploidy of the pathogen; the current pipeline is suited to analyze haploid pathogens.
- The genetic diversity of the pathogen; the pipeline compares vcf files generated from mapping reads on the same reference; a complete change to the approach would be required in cases where vcf files could not be produced as such. When this approach is possible, a higher genetic distance between the viruses of a co-infection would facilitate its detection.
- A pre-existing clade/lineage definition; although, the database approach could be adapted to create placeholder “clades” of sufficiently distant viruses when these pre-existing definitions don’t exist.
- A well curated database of fasta sequences of the pathogen, containing currently circulating viruses.

Therefore, this approach could easily be applicable to flu (HxNx types analyzed separately), monkeypox or respiratory syncytial viruses for example. It is currently in development for flu viruses, with promising early results. It will be developed progressively for any viruses analyzed in our laboratory, and will be released publicly on our github repository when ready.

The manuscript discussion has been updated to address this aspect of the subject (lines 400-408).

Q14. Lines 440-442 – are these colours meant to be included in the legend for figure 3? They don't seem to correspond to colours in that figure?

A14. Thank you for pointing out this error. The legend has been updated to cite only the colors included in Figure 3.

Q15. Figure 1A – I may be missing this but the ratio of Delta-specific mutations and the ratio of Omicron-specific mutations don't seem to be described anywhere which makes it difficult to understand exactly what is being plotted in this figure. It would be useful to describe what these terms mean.

A15. We apologize if our wording was not clear. We have now replaced “ratio of Delta-specific mutations” and “ratio of Omicron-specific mutations” by “detection rate of Delta-specific mutations” and “detection rate of Omicron-specific mutations”, respectively, to better match with the text. These detection rates are defined as the number of Delta- or Omicron-specific variants found in each sample (as minor or major allele) out of the total number of Delta- or Omicron-specific variants based on covariants.org. The definition has been added in the legend (lines 545-549).

Reviewer #3 (Remarks to the Author):

The manuscript entitled "Detection and prevalence of SARS-CoV-2 co-infections during the Omicron variant circulation, France, December 2021 - February 2022" by Bal et al. describes an automated data analysis pipeline for amplicon sequencing-based identification of co-infection with 2 of the 5 known variants of concern of SARS-CoV-2. The authors describe the publicly-available pipeline and validate its performance on a titrated mix of cultured Delta:Omicron variants. They then apply the pipeline to amplicon sequencing data from >20,000 patient samples positive for SARS-CoV-2 during December 2021 - February 2022 and identify 53 reproducibly detectable co-infections. The authors discuss the importance of identifying co-infections in an automated and scalable way, including to identify recombination events and limit description of artificially chimeric viral consensus genome sequences.

Major comments

Q1a. Line 190: "Importantly, consensus sequence calling based on majority rule resulted in artefactual chimeric Delta-Omicron sequences for several mixes and with different patterns depending on the primers used for amplification (Fig S1)" -> This is an important observation. Is this also true for other commonly used pipelines? Can the authors test?

A1a. As suggested by the reviewer, we have tested two other commonly used pipelines, namely the DRAGEN pipeline (Illumina) recommended with the COVIDSeq kit (Illumina) and the Connor lab pipeline (named "IVAR" in **Fig S5**), one of the first pipelines published to analyze SARS-CoV-2 amplicon sequences (<https://github.com/connor-lab/ncov2019-artic-nf>) which was used by the COG-UG. Consensus sequences called by these two pipelines also included chimeric sequences for Delta:Omicron mixes with 10-50% Omicron (**Fig S5**).

These new results are presented in a new supplementary figure (**Fig S5**) and in the result section (lines 226-232) reproduced below:

"Sequences bearing both Delta- and Omicron-specific polymorphisms were found independently of the bioinformatic pipeline used to call the consensus sequence (**Fig S5**). Omicron sequences bearing Delta-specific mutations were found in mixes with Delta expected frequency of 10 to 30%. The highest number of Delta-specific mutations was observed in the 20:80 mixes sequenced with Midnight primers, and in the 30:70

mixes sequenced with Artic primers (**Fig S5**), which were the mixes with 50% measured frequency of Delta (**Fig 1B**).”

Q1.b. Did the authors quantify how often this was the case with natural co-infections?

A1.b. Thank you for this suggestion. We have performed additional analysis to quantify the chimeric sequences in natural co-infections that are now presented in the main results (lines 293-297) and in **Fig 3C**.

The number of mutations specific to the minor lineage present in the consensus sequence were used as a proxy to identify chimeric sequences (**Fig 3C**). Chimeric sequences were identified in 69/106 (65%) natural co-infected duplicates. All samples with a minor lineage above 38% were chimeric with a median of 4 minor lineage specific mutations present in the consensus sequence (**Fig 3C**).

Q2. Line 205: “A secondary lineage is identified only if 6 specific mutations are present” -> How was this cutoff derived and optimized? Will this perform equally well for lineages with fewer lineage-specific mutations? Can the authors provide data?

A2. As suggested, we have added 3 sections in supplementary methods and results to explain how the 6 specific mutation cutoff was derived, its impact on the detection of the different lineages, and on our results. These data are supported by 4 new supplementary figures (**FigS1, S2, S3 and S7**) and are referenced in the main text (lines 159-164 and 277-279).

These supplementary data are reproduced below:

Determination of the optimal cutoff value for detecting the secondary lineage

The appropriate cutoff was determined empirically while training the pipeline on real data and positive controls included in each run. Based on the artificial Delta:Omicron mixes and all positive controls (cell culture isolates) sequenced during the study period (n=1084, including 892 pure cell culture isolates and 192 experimental mixes) a ROC curve was generated with the Youden index indicating a putative optimal point for this cutoff (**Fig S1A**). While the Youden index’s best score corresponds to a minimum of 4

lineage-specific variants cutoff (TPR-FPR=0.952), the chosen cutoff of 6 while designing the study had a close score (TPR-FPR=0.932) and permitted the exclusion of false positives identified in the real dataset. Indeed, out of 7 samples which had 4 or 5 secondary lineage-specific mutations at first passage, only 1 sample retained 5 secondary lineage-specific mutations at second passage (**Fig S2**). Moreover, in the real dataset, among samples with a non null secondary lineage-specific mutations count, 2143/2257 (94.9%) samples had a mutation count ranging from 1 to 5 as shown in **Fig S1B**, which was similar to the proportion of 454/477 (95,1%) found in the positive controls sequenced during the study period.

Impact of the cutoff value on different lineages detection

The number of defining mutations depends on the lineage with a median of 40 defining mutations per lineage, a minimum of 26 mutations for AY.9.2 and a maximum of 64 mutations for BA.2 (**Fig S3A**). For the main lineage definition, the 6 defining mutations threshold is very low compared to the number of defining mutations per lineage (**Fig S3A**). On the other hand, the 6 specific mutations threshold to find a secondary lineage limits our ability to detect co-infections between closely related lineages. Specifically, 12321/43472 (28%) Delta/Delta co-infections with different lineages cannot be detected using the 6 specific mutation threshold (**Fig S3B**). At the time of writing, there was only one sublineage of BA.1 defined, BA.1.1, and co-infections between BA.1 and BA.1.1 would not be detected either as BA.1.1 has only 3 specific mutations in addition to the defining mutations of BA.1.

Impact of the cutoff value on the prevalence of natural co-infections

The number of samples with a secondary lineage identified in their first replicate depends on the cutoff value used to determine the secondary lineage (**Fig S7**). The

chosen cutoff was 6 specific mutations, resulting in 0.29% of potentially co-infected samples (95% CI: 0.22%-0.37%, assuming a binomial distribution). Using only 1 specific mutation as the cutoff would identify a potential co-infection in 10.3% of the samples. In contrast, a cutoff between 3 and 17 specific mutations would not significantly change the estimated prevalence of co-infection (Fisher's p-value >0.05) (**Fig S7**). Excluding 1 specific mutation, the estimated prevalence varied between 0.542% (for a cutoff of 2 specific mutations) and 0.00935% (for a cutoff of 44 defining mutations).

Additionally, users can easily change this cut-off to match their experimental settings (contamination background may vary depending on experimental procedures and automation) and when seeking co-infection by closer lineages. This consideration has been added in the main text (lines 404-408).

Q3. Line 225: "To rule out potential contamination during initial sequencing, all these 64 samples were re-extracted and sequenced in duplicate. In total, 53 samples had a positive secondary lineage mutation ratio in duplicate" -> Can the authors comment on results for the remaining 11 samples?

A3. Among the 11 samples, 3 samples were mistakenly included in the initial version of the manuscript as they did not reach the 90% genome coverage threshold in the second passage and are therefore not part of the 21,387 samples with WGS analyzed in the manuscript. Only 61 samples had >90% genome coverage in both duplicates, including 8 that had discordant secondary lineage mutation ratios.

Additional analyses were performed for these 8 samples (**Fig S8** and **Fig S9**) and are now described in the **Supplementary results** reproduced below:

Investigation of samples with discordant secondary lineage mutation ratios in duplicates

Among 61 samples with positive secondary lineage mutation ratios in the first passage, 8 samples had null ratios in the second passage, thus no longer found to harbour co-

infections. These samples were characterized by lower secondary lineage mutation ratios in the first replicate than confirmed co-infected samples (median ratio of 0.38 vs 0.78, Kruskal-Wallis p-value = 0.0083) (**Fig S8A**) and lower relative abundance of the minor lineage in the first replicate (median relative abundance of 9.5% vs 18%, Kruskal-Wallis p-value = 0.03) (**Fig S8 B**). The distribution of minor alleles along the genome in duplicate suggests that a contamination arose during the first sequencing process, either during RNA extraction, PCR amplification or library preparation (**Fig S9**). For $\frac{5}{8}$ samples, minority variants from the secondary lineage were present all along the genome, suggesting that the contamination arose most likely during the RNA extraction. For $\frac{3}{8}$ samples, minority variants were present only on 2 to 3 amplicons, suggesting contamination may have arisen after PCR amplification with a limited number of amplicons. Additionally, for 4 samples, as secondary lineage-specific mutation relative frequencies were $\sim 5\%$, we cannot rule out that a co-infection could have been present, but allele frequencies were not reproducible around 5%.

These results are referenced in the main text (lines 280-283) and detailed in the **Supplementary results, Fig S8** and **Fig S9**.

Q4. Line 233: “Uniform frequencies of lineage-specific mutations” -> How was “uniform” defined and the determination made? Solely by manual inspection? Could the authors include a flag in the software or at least discuss how this could be done?

A4. The two recombinant SARS-CoV-2 were initially suspected by visual inspection. As suggested, we assessed whether these recombinants may be detected using a numerical score. This statistical approach allowed us to identify a third potential recombinant, as described in the main text (lines 301-306).

The statistical approach is described in **Fig S13** and in **Supplementary results** reproduced below:

Evaluation of uniform frequencies among co-infected samples for screening of recombinant events

To assess whether recombinants may be present in co-infected samples, the cumulative sum of allele frequencies were used. We considered that when two specific lineages are present in different proportions, the cumulative variant allele frequencies of the specific defining-lineage positions should follow a linear distribution. In addition, we considered that the linear distribution of each lineage should have a 0 intercept. If a recombinant is present, we would observe a break in uniformity of specific positions frequencies and thus a default in linear regression and a deviation from 0 intercept. From these hypotheses, we computed the adjusted r-squared value from the regression model of each major or minor lineage identified in samples and the p-value considering a 0 intercept under null hypothesis. To note, we removed nucleotide positions known to be associated with Artic primer bias regardless of the version (amplicons 76 and 88-90).

Adjusted r-squared of linear models from mixes were all above 0.98 (0.988-0.999). According to these results (**Fig S13**), we considered co-infected samples with an r-squared adjusted score under 0.98. Three samples were detected with consistent results in duplicate sequencing (021228537801; 0212296567 and 722000801801). Additionally, 5 samples had an adjusted R-squared below 0.98, but the uniformity of either major or minor lineage-specific frequencies was disrupted by sporadic positions. The pattern of these 5 samples is likely due to sequencing competition, whereas recombinant patterns follow a disruption in frequency uniformity affecting several and consecutive positions on different amplicons. By combining a threshold of adjusted R squared below 0.98 and a p.value below 0.05 testing 0 intercept, 3 samples were rejected and therefore suspected to be recombinant. This statistical approach within our pipeline is interesting for warning about possible recombinants. Nonetheless, visual inspection

remains essential, as well as other specific investigations such as phasing and culture approaches to prove the presence of a recombinant among viral sub-populations.

Q5. Do the authors have access to information on the immune status of patients with co-infections?

A5. The vaccination status of the co-infected patients was retrieved from national health insurance agency files.

Among the 53 co-infected patients, a total of 21 patients (39.6%) were not vaccinated, corresponding to 9/16 (56.3%) hospitalized patients and 12/36 (33.3%) outpatients.

A total of 32 patients (60.4%) were vaccinated (1 with one dose, 18 with 2 doses, and 13 with 3 doses). The median (IQR) delay between last dose and the date of PCR test was 124 (57-166) days.

In France, the overall vaccination coverage during this period was above 90%. A strict comparison with the vaccination coverage of co-infected patients could not be performed as it required to stratify the vaccination coverage by clinical status which was not available for all patients at national level.

These new results have been added in the abstract (lines 36-37) and results section (lines 340-345) and are discussed (lines 413-417).

Q6a. Fig 3: can the authors add a panel showing the distribution of median allele frequencies for the ‘secondary virus’ in natural co-infections?

A6a. Thank you for this suggestion. We have added a new panel **B** in **Fig 3** as suggested to show the distribution of the least abundant virus (called “minor virus”) in natural co-infections. Please note that we chose to plot the frequency of the “minor virus” rather than the “secondary virus” as this latter is not always the least abundant. Indeed, the secondary lineage is chosen based on the ratio of found/expected specific mutations and not on relative frequency.

These results are now described in the main text (lines 287-292) reproduced below:

For these 53 natural co-infections, relative abundance of minor lineages were correlated in duplicate (Pearson correlation coefficient = 0.92, p-value < 2.2e-16) with a median relative abundance of 20% (iqr=26.25) (**Fig 3B**). Minor lineages were identified as the

secondary lineages in both duplicates for 45 samples but were identified either as the main or secondary lineage in duplicates for 8 samples with a higher relative abundance of minor lineage (median relative abundance = 38,75%, Kruskal-Wallis p-value = 9.479e-06) (**Fig 3B**).

Q6b. Using the information from Fig 1B, can the authors calculate the corrected variant ratio in the natural co-infection cases (Delta- Omicron)?

A6b. **Fig 1B** reports the relationship between measured vs expected frequency of Delta in experimental mixes of BA.1 and B.1.617.2 viral isolates. This relationship could be used to correct the relative abundance of lineages in natural co-infections as suggested, but only in those co-infected with BA.1 and B.1.617.2 lineage. Indeed, specific mutations are defined based on each lineage by seqmet. As only a minority of natural co-infections (10/53, 19%) were identified as co-infections between BA.1 and B.1.617.2 (**Fig S10**), we did not perform this correction and chose to present in **Fig 3B** the measured relative abundance of minor lineage for all co-infected samples.

Q6c. What was the rate of chimeric consensus sequences in natural co-infections?

A6c. We have performed additional analysis to determine the rate of chimeric sequences in natural co-infections as suggested. The number of mutations specific to the minor lineage present in the consensus sequence was used as a proxy to identify chimeric sequences (**Fig 3C**). Chimeric sequences were identified in 69/106 (65%) natural co-infected duplicates. All samples with a minor lineage above 38% were chimeric with a median of 4 minor lineage-specific mutations present in the consensus sequence (**Fig 3C**).

These new results are presented in the main text (lines 293-297) and in a new panel Fig 3 (**Fig 3C**).

Minor comments

Q7. Can the authors provide viral loads in copies rather than Ct? e.g., line 173 “Resulting mixes had fixed viral loads (median Ct = 22.8; IQR=1.5)”

A7. Quantitative viral load was determined for each mix with RT-qPCR SARS-CoV-2 R-gene kit (bioMérieux, Lyon, France) using four quantification standards targeting the SARS-CoV-2

N gene:QS1 to QS4 respectively $2.5 \cdot 10^6$, $2.5 \cdot 10^5$, $2.5 \cdot 10^4$, $2.5 \cdot 10^3$ copies/mL of a SARS-CoV-2 DNA standard. The corresponding median viral load was 4.2 log₁₀ cp/ml; IQR= 0.4.

We added these new data in the methods and results sections (lines 113-117 and 204-205) in the revised version of the manuscript. Viral loads for each mix are now included in **Table S3** and **S4**.

Q8. Line 214: “Secondary lineage mutation ratios were between 0.216 and 0.941 (Fig 2B)”; This seems surprising given the high depth of coverage (Line 176: “All mixes were sequenced to 1 M paired-end reads leading to SARS-CoV-2 genome covered > 98% with median coverage of 2276X (IQR=315X)” -> Can the authors discuss the low ratios for some of the cases?

A8. In mixes with 10% expected frequency of Omicron, most mutations were missed because they did not reach 5 % relative frequency (threshold used in seqmet to call a variant) (**Table S3**). In addition, some mutations were missed in the other mixes with a primer-specific pattern. Omicron-specific mutations at positions 22813 (S:K417N), 22882 (S:N440K), 22898 (S:G446S) were missed with Artic V4 primers, while Delta-specific mutation at position 22917 (S:L452R) was amplified preferentially in Delta:Omicron mixes, in relation with amplicon 76 (nt 22677- 23028) drop-out in Omicron samples (<https://community.artic.network/t/sars-cov-2-v4-1-update-for-omicron-variant/342>). Artic V4 amplicon 89 (nt 26587-26956) also led to Delta preferential amplification: Omicron-specific mutation at position 26709 (M:A63T) was missed in mixes with up to 60% Omicron expected frequency, while Delta-specific mutation at position 26767 (M:I82T) was preferentially sequenced. Mutations comprised in the Midnight amplicons 24 (nt 23544 –24714) and 28 (nt 27808 – 28985) also had amplification bias with preferential amplifications of Delta over Omicron, as reported in Rockett et al, Nature Com, 2022. Omicron-specific mutations at positions 23854 (S:N764K), 23948 (S:D796Y), 24130 (S:N856K), 24424 (S:Q954H), 24469 (S:N969K), 24503 (S:L981F), 28311 (N:P13L) and 28361 (N:30Del) were missed in several low frequency Omicron mixes sequenced with midnight primers, while Delta-specific mutations at positions 27874 (non coding), 28247 (ORF8: 118Del), 28270 (non coding) and 28461 (N:D63G) were present at high relative frequency.

In mixes with a low expected frequency of Delta (10-20% for Midnight primers; 20-30% for Artic primers), these amplification biases resulted in chimeric sequences characterized by Omicron sequences bearing Delta-specific mutations in these regions.

We have now included these data in **Table S3** and discuss these results in the **Supplementary results** (section: *Primer bias*) and in the main text (lines 260-262).

Q9. Line 224: “Among the 21,387 samples, 64 samples (0.30%) had a secondary lineage identified with positive ratios (Fig 3)” -> What is the definition of ‘positive ratio’?

A9. We mistakenly used “positive ratios” in place of “positive secondary lineage mutation ratio” (defined in the manuscript lines 155-156). Thank you for pointing this mistake out. It has been corrected in the revised version of the manuscript.

Q10. Line 143: “For Delta:Omicron mixes, vcf files were also analyzed using the curated list of clade-defining mutations of Delta and Omicron (lineage BA.1) as previously published.” -> Can the authors provide these results?

A10. We apologize if our wording was not clear. These results were presented in the first paragraph of the result section “Evaluation of 4 different primer sets to detect SARS-CoV-2 co-infections using WGS”. We have added a sentence to clarify this (lines 172-173).

Q11. Fig 1A/B and 2A/B -> The red-blue color gradient is lacking resolution making it hard to map colors to ratios of the virus mix-> Suggest 3-color gradient

A11. We have updated Fig 1AB and Fig 2AB to set a 3-color gradient as suggested.

REVIEWERS' COMMENTS

Reviewer #2 (Remarks to the Author):

The Authors have responded to my comments very thoroughly. I have no further comments.

I've had a look through the responses to comments from the other 2 reviewers. The Authors seem to have done a really thorough job responding to everything. All of the new/updated analyses and text appear to be well supported and well described. I only have a few very minor comments:
Reviewer #1 Q4 - in the new figure S8, it could be clearer that the points are the 8 samples that don't show coinfection in both extractions. This could be described specifically in the legend and "Natural coinfection" in the figure might be renamed to make clear that the shape distinguishes samples supported/not supported in both extractions

Reviewer #1 Q5 - is it possible to give alias names to each of the potentially recombinant samples? As a reader it is difficult to distinguish 021228537801 from 021229656701 and match these with the figures. A short name for each sample would greatly help to read and interpret figure S14

Reviewer #1 Q7 - I can't see the mutation names in Fig S4. Are they the text next to each point? While it's possible to zoom into this figure, it's not possible to zoom in far enough to see this text (on my computer at least). Are the mutation names necessary?

Reviewer #2 (Remarks to the Author):

The Authors have responded to my comments very thoroughly. I have no further comments. I've had a look through the responses to comments from the other 2 reviewers. The Authors seem to have done a really thorough job responding to everything. All of the new/updated analyses and text appear to be well supported and well described.

We thank the reviewer for these positive comments.

I only have a few very minor comments:

Reviewer #1 Q4 - in the new figure S8, it could be clearer that the points are the 8 samples that don't show coinfection in both extractions. This could be described specifically in the legend and "Natural coinfection" in the figure might be renamed to make clear that the shape distinguishes samples supported/not supported in both extractions

Answer – We have renamed “Natural coinfection” in “Confirmed coinfection in duplicates” in the revised version of the S8 figure. It is also indicated in the legend:

“Samples found to harbor a coinfection (i.e. which have positive secondary lineage mutation ratios in duplicate) are depicted as triangles, while samples with discordant secondary lineage mutation ratios are depicted as dots »

Reviewer #1 Q5 - is it possible to give alias names to each of the potentially recombinant samples? As a reader it is difficult to distinguish 021228537801 from 021229656701 and match these with the figures. A short name for each sample would greatly help to read and interpret figure S14

Answer- Alias names were attributed for each potentially recombinant samples as suggested. The figure S14 was updated accordingly

Reviewer #1 Q7 - I can't see the mutation names in Fig S4. Are they the text next to each point? While it's possible to zoom into this figure, it's not possible to zoom in far enough to see this text (on my computer at least). Are the mutation names necessary?

Answer- After downloading the manuscript and opening with adobe, it is possible to zoom in far enough to see the mutation names. We think that this information is useful, notably for identifying chimeric mutations and we suggest to keep the mutations names in this figure.